# Convex Formulations for Training Two-Layer ReLU Neural Networks

**Karthik Prakhya**[*]**, Tolga Birdal**[†] **& Alp Yurtsever**[*]
[*] Department of Mathematics and Mathematical Statistics, Umeå University, Sweden
[†] Department of Computing, Imperial College London, United Kingdom

## Abstract

Solving non-convex, NP-hard optimization problems is crucial for training machine learning models, including neural networks. However, non-convexity often leads to black-box machine learning models with unclear inner workings. While convex formulations have been used for verifying neural network robustness, their application to training neural networks remains less explored. In response to this challenge, we reformulate the problem of training infinite-width two-layer ReLU networks as a convex completely positive program in a finite-dimensional (lifted) space. Despite the convexity, solving this problem remains NP-hard due to the complete positivity constraint. To overcome this challenge, we introduce a semidefinite relaxation that can be solved in polynomial time. We then experimentally evaluate the tightness of this relaxation, demonstrating its competitive performance in test accuracy across a range of classification tasks.

## 1 Introduction

The outstanding performance of deep neural networks has driven significant changes in the research directions of optimization for machine learning over the past decade, replacing traditional convex optimization techniques with non-convex methods. Nonetheless, non-convexity introduces significant challenges to the analysis, resulting in the use of models that lack a comprehensive explanation of their inner workings. While convex formulations have been studied for enhancing reliability through applications like estimating the Lipschitz constant of neural networks (Fazlyab et al., 2019; Chen et al., 2020; Latorre et al., 2020; Pauli et al., 2023) or verifying their robustness (Raghunathan et al., 2018; Fazlyab et al., 2020; Zhang, 2020; Lan et al., 2022; Chiu & Zhang, 2023), their application to the training of neural networks remains less explored.

In this paper, we study convex optimization representations for training a two-layer neural network with rectifier linear unit (ReLU) activations and a sufficiently wide hidden layer. While the relationships between classical matrix factorization problems and two-layer linear neural networks, as well as those between matrix factorization and semidefinite programming, are well-established, our understanding of the connections between neural networks and convex optimization becomes less clear when non-linear activation functions are involved. Our paper advances these connections by establishing links between convex optimization and neural networks with ReLU activations.

In light of this brief introduction, we summarize our key contributions:

- We present a novel copositive program for training a two-layer ReLU neural network, providing an exact reformulation of the training problem when the network is sufficiently wide. Specifically, for fixed input and output dimensions and a given number of datapoints, there exists a finite critical width beyond which the network's expressivity saturates, making the training problem equivalent to the proposed copositive formulation. We further extend this equivalence to the more general infinite-width regime, where the neural network is represented as a Lebesgue integral over a probability measure on the weights.
- Although copositive programs are convex, solving them is NP-hard (Bomze et al., 2000). To mitigate this obstacle, we propose a semidefinite programming relaxation of the original formulation. We then numerically evaluate its tightness on two synthetic examples. We also assess its performance on real-data classification tasks, where, combined with a rounding heuristic, our approach achieves competitive test accuracy compared to Neural Network

Gaussian Process (NNGP) (Lee et al., 2018) and Neural Tangent Kernel (NTK) (Jacot et al., 2018) methods.

We make our implementation available under https://github.com/KarthikPrakhya/SDPNN-IW.

## 2 BACKGROUND

**Definition 1** (Positive semidefinite cone). *A symmetric matrix $\boldsymbol{W} \in \mathbb{R}^{n \times n}$ is said to be positive semidefinite if $\boldsymbol{u}^\top \boldsymbol{W} \boldsymbol{u} \geq 0$ for all $\boldsymbol{u} \in \mathbb{R}^n$. The set of all positive semidefinite matrices forms a self-dual convex cone, known as the positive semidefinite cone, which can be defined by*

$$\mathcal{PSD}_n := \operatorname{conv}\{\boldsymbol{w}\boldsymbol{w}^\top : \boldsymbol{w} \in \mathbb{R}^n\}, \tag{1}$$

*where* conv *represents the convex hull. We omit the subscript when the size is clear from the context.*

**Semidefinite programming**. (SDP) is a powerful framework in convex optimization focused on minimization of a convex objective over the positive semidefinite cone. SDPs can be solved in polynomial time under mild technical assumptions using interior-point methods (Vandenberghe & Boyd, 1996). We refer to (Majumdar et al., 2020; Yurtsever et al., 2021b) for an overview of recent advances in more scalable SDP solvers.

SDPs have been applied in neural networks for various tasks (often as a convex relaxation), including the estimation of Lipschitz constant (Chen et al., 2020; Fazlyab et al., 2019; Latorre et al., 2020; Pauli et al., 2023), verification of neural networks (Raghunathan et al., 2018; Zhang, 2020; Chiu & Zhang, 2023; Lan et al., 2022; Fazlyab et al., 2020), and for stability guarantees (Pauli et al., 2022; 2021; Revay et al., 2020; Yin et al., 2021). A recent line of work, initiated by Pilanci & Ergen (2020), investigates the use of SDPs for training neural networks. We provide a detailed discussion and comparison with these approaches in Section 6.

**Definition 2** (Copositive cone). *A matrix $\boldsymbol{W} \in \mathbb{R}^{n \times n}$ is said to be copositive if $\boldsymbol{u}^\top \boldsymbol{W} \boldsymbol{u} \geq 0$ for all $\boldsymbol{u} \in \mathbb{R}^n_+$. The cone of copositive matrices is called the copositive cone ($\mathcal{COP}$).*

**Definition 3** (Completely positive cone). *The dual of copositive cone is known as the completely positive cone, defined as*

$$\mathcal{CP}_n := \operatorname{conv}\{\boldsymbol{w}\boldsymbol{w}^\top : \boldsymbol{w} \in \mathbb{R}^n_+\}. \tag{2}$$

*It is easy to see that completely positive cone is a subset of positive semidefinite cone, and positive semidefinite cone is a subset of copositive cone, as shown in Figure 1.*

**Copositive programming** (CP) is a subfield of convex optimization concerned with minimization of a convex objective function over copositive or completely positive matrices. Despite its convexity, solving a CP is NP-Hard (Bomze et al., 2000). Numerous NP-Hard problems are formulated as a CP, including the binary quadratic problems (Burer, 2009), problems of finding stability and chromatic numbers of a graph (De Klerk & Pasechnik, 2002; Dukanovic & Rendl, 2010), 3-partitioning problem (Povh & Rendl, 2007), and the quadratic assignment problem (Povh & Rendl, 2009). We refer to the excellent surveys (Dür, 2010; Dür & Rendl, 2021) and references therein for more examples.

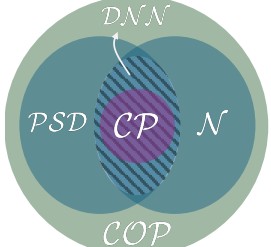

Figure 1: Cones.

Recent research has broadened the scope of CP into data science. For example, CP formulations have been utilized in machine learning for verifying neural networks (Brown et al., 2022), and problems such as graph matching and permutation synchronization in computer vision (Yurtsever et al., 2022). CP formulations have also been proposed for training two-layer ReLU networks, though under specific data-related assumptions (Sahiner et al., 2021).

**Definition 4** (Non-negative cone). *The non-negative cone $\mathcal{N}$ comprises of entry-wise non-negative matrices.*

**Definition 5** (Doubly non-negative cone). *The doubly non-negative cone is the set of matrices that are both positive semidefinite and elementwise non-negative, defined by $\mathcal{DNN} := \mathcal{PSD} \cap \mathcal{N}$. It is an important subset of the positive semidefinite cone that contains the cone of completely positive matrices and is frequently used in relaxations of CP formulations.*

**Definition 6** (Rank). *For a matrix $\boldsymbol{W} \in \mathcal{PSD}_n$, the **rank** is the minimum number of vectors $\boldsymbol{w}_i \in \mathbb{R}^n$ required to express $\boldsymbol{W}$ as a convex combination of $\boldsymbol{w}_i \boldsymbol{w}_i^\top$.*

**Definition 7** (CP-rank). *We refer to the rank restricted to decompositions involving non-negative vectors as the CP-rank: The **CP-rank** of a matrix $\boldsymbol{W} \in \mathcal{CP}_n$ is the minimum number of **non-negative vectors** $\boldsymbol{w}_i \in \mathbb{R}_+^n$ that are needed to express $\boldsymbol{W}$ as a convex combination of $\boldsymbol{w}_i \boldsymbol{w}_i^\top$.*

It is well known that the rank of any $n \times n$ matrix $\boldsymbol{W}$ cannot exceed $n$. The following result from (Barioli & Berman, 2003, Theorem 3.1) characterizes the maximal CP-rank.

**Lemma 1.** *Let $\boldsymbol{W}$ be a completely positive matrix of rank $r \geq 2$. Then, the CP-rank of $\boldsymbol{W}$ cannot exceed $r(r+1)/2 - 1$. As an immediate corollary, the CP-rank of any $n \times n$ completely positive matrix for $n \geq 2$ is bounded above by $n(n+1)/2 - 1$, which we denote as the maximal CP-rank.*

## 3 CP FORMULATION FOR TRAINING TWO-LAYER ReLU NETWORKS

We now present our CP formulation for training two-layer ReLU networks. We start by the *wide* regime with finite width before moving onto the infinite case that carries theoretical importance.

**Definition 8** (Two-Layer ReLU Network). *A two layer neural network $\psi : \mathbb{R}^d \to \mathbb{R}^c$ with $m$ hidden neurons and ReLU activations is given by*

$$\psi(\boldsymbol{x}) = \sum_{j=1}^m (\boldsymbol{x}^\top \boldsymbol{u}_j)_+ \boldsymbol{v}_j^\top, \tag{NN}$$

*where the function $(\cdot)_+ = \max(0, \cdot)$ denotes the ReLU activation, $\{\boldsymbol{u}_j \in \mathbb{R}^d\}_{j=1}^m$ are the weights for the first layer and $\{\boldsymbol{v}_j \in \mathbb{R}^c\}_{j=1}^m$ are the second layer weights.*

Assuming that the data matrix $\boldsymbol{X} \in \mathbb{R}^{n \times d}$ and labels $\boldsymbol{Y} \in \mathbb{R}^{n \times c}$ are given, and using the mean squared error loss and Tikhonov regularization (also known as weight-decay), the problem for training this neural network can be formulated as follows:

$$\min_{\substack{\boldsymbol{u}_j \in \mathbb{R}^d \\ \boldsymbol{v}_j \in \mathbb{R}^c}} \left\| \sum_{j=1}^m (\boldsymbol{X}\boldsymbol{u}_j)_+ \boldsymbol{v}_j^\top - \boldsymbol{Y} \right\|_F^2 + \frac{\gamma}{2} \sum_{j=1}^m (\|\boldsymbol{u}_j\|_2^2 + \|\boldsymbol{v}_j\|_2^2), \tag{NN-Train}$$

where $\gamma \geq 0$ is the regularization parameter.

In the following, our main result is an equivalence between the above training problem and the following convex optimization problem with $\mathcal{PSD}$ and $\mathcal{CP}$ constraints when $m$ is sufficiently large:

**Theorem 1** (CP Formulation). *Consider the two-layer ReLU neural network in Eq. (NN). For any fixed input dimension $d$, output dimension $c$, and number of data points $n$, there exists a finite critical width $R \leq \max\{p, 2n^2 + n - 1\}$, where $p = c + d + 2n$, such that, for $m \geq R$ hidden neurons, the training problem (NN-Train) is equivalent to the convex optimization problem (CP-NN), meaning that they have identical optimal values:*

$$\min_{\boldsymbol{\Lambda} \in \mathbb{R}^{p \times p}} \quad \left\| \boldsymbol{P}_\alpha \boldsymbol{\Lambda} \boldsymbol{P}_v^\top - \boldsymbol{Y} \right\|_F^2 + \frac{\gamma}{2} \left( \operatorname{trace}(\boldsymbol{P}_u \boldsymbol{\Lambda} \boldsymbol{P}_u^\top) + \operatorname{trace}(\boldsymbol{P}_v \boldsymbol{\Lambda} \boldsymbol{P}_v^\top) \right)$$
$$\text{s.t.} \quad \operatorname{trace}(\boldsymbol{P}_\alpha \boldsymbol{\Lambda} \boldsymbol{P}_\beta^\top) + \operatorname{trace}(\boldsymbol{M} \boldsymbol{\Lambda} \boldsymbol{M}^\top) = 0 \tag{CP-NN}$$
$$\boldsymbol{\Lambda} \in \mathcal{PSD}$$
$$\boldsymbol{P}_{\alpha\beta} \boldsymbol{\Lambda} \boldsymbol{P}_{\alpha\beta}^\top \in \mathcal{CP},$$

*where $\boldsymbol{M} = -\boldsymbol{P}_\alpha + \boldsymbol{P}_\beta + \boldsymbol{X}\boldsymbol{P}_u$ and the selection matrices are defined as follows:*

$$\boldsymbol{P}_u = [\boldsymbol{0}_{d \times n} \quad \boldsymbol{0}_{d \times n} \quad \boldsymbol{I}_{d \times d} \quad \boldsymbol{0}_{d \times c}] \quad \boldsymbol{P}_\alpha = [\boldsymbol{I}_{n \times n} \quad \boldsymbol{0}_{n \times n} \quad \boldsymbol{0}_{n \times d} \quad \boldsymbol{0}_{n \times c}]$$
$$\boldsymbol{P}_v = [\boldsymbol{0}_{c \times n} \quad \boldsymbol{0}_{c \times n} \quad \boldsymbol{0}_{c \times d} \quad \boldsymbol{I}_{c \times c}] \quad \boldsymbol{P}_\beta = [\boldsymbol{0}_{n \times n} \quad \boldsymbol{I}_{n \times n} \quad \boldsymbol{0}_{n \times d} \quad \boldsymbol{0}_{n \times c}] \quad \text{and} \quad \boldsymbol{P}_{\alpha\beta} = \begin{bmatrix} \boldsymbol{P}_\alpha \\ \boldsymbol{P}_\beta \end{bmatrix}$$

It is noteworthy that this formulation is independent of $m$ and hence does not scale in size with the number of hidden neurons. Our analysis is also independent of the loss. Hence, we can get the same guarantees for any convex loss function as long as we use the corresponding loss function in (CP-NN) as well. Additional bias terms can be incorporated into this formulation by augmenting the input data with a column of ones.

*Proof sketch.* The first challenge in (NN-Train) problem arises from the non-linear ReLU activation function, which we address by decomposing $\boldsymbol{X}\boldsymbol{u}$ into its positive and negative parts, inspired by (Brown et al., 2022). Specifically, for non-negative vectors $\boldsymbol{\alpha}$ and $\boldsymbol{\beta}$ such that $\mathrm{trace}(\boldsymbol{\alpha}\boldsymbol{\beta}^\top) = 0$, we have $\boldsymbol{X}\boldsymbol{u} = \boldsymbol{\alpha} - \boldsymbol{\beta}$ if and only if $\boldsymbol{\alpha} = (\boldsymbol{X}\boldsymbol{u})_+$ and $\boldsymbol{\beta} = (\boldsymbol{X}\boldsymbol{u})_+ - \boldsymbol{X}\boldsymbol{u}$, corresponding to the positive and negative components, respectively.

Introducing the variable $\boldsymbol{\lambda}_j = [\boldsymbol{\alpha}_j^\top\ \boldsymbol{\beta}_j^\top\ \boldsymbol{u}_j^\top\ \boldsymbol{v}_j^\top]^\top \in \mathbb{R}^p$, we can reformulate (NN-Train) as:

$$\min_{\boldsymbol{\lambda}_j \in \mathbb{R}^p} \quad \left\| \sum_{j=1}^m \boldsymbol{P}_\alpha \boldsymbol{\lambda}_j \boldsymbol{\lambda}_j^\top \boldsymbol{P}_v^\top - \boldsymbol{Y} \right\|_F^2 + \frac{\gamma}{2} \sum_{j=1}^m \left( \|\boldsymbol{P}_u \boldsymbol{\lambda}_j\|_2^2 + \|\boldsymbol{P}_v \boldsymbol{\lambda}_j\|_2^2 \right), \tag{3}$$

$$\text{s.t.} \quad \boldsymbol{P}_{\alpha\beta}\boldsymbol{\lambda}_j \geq \boldsymbol{0}_{2n\times 1}, \quad \mathrm{trace}(\boldsymbol{P}_\alpha \boldsymbol{\lambda}_j \boldsymbol{\lambda}_j^\top \boldsymbol{P}_\beta^\top) = 0, \quad \boldsymbol{M}\boldsymbol{\lambda}_j = \boldsymbol{0}_{n\times 1}.$$

Our goal is to show that the problem (CP-NN) is equivalent to problem (3) when $m$ is sufficiently large. To this end, we establish that any matrix $\boldsymbol{\Lambda} \in \mathcal{PSD}$ satisfying $\boldsymbol{P}_{\alpha\beta}\boldsymbol{\Lambda}\boldsymbol{P}_{\alpha\beta}^\top \in \mathcal{CP}$ can be factorized as $\boldsymbol{\Lambda} = \sum_{j=1}^R \boldsymbol{\lambda}_j \boldsymbol{\lambda}_j^\top$, where $\boldsymbol{P}_{\alpha\beta}\boldsymbol{\lambda}_j \geq \boldsymbol{0}$, and $R$ is a finite number bounded by $R \leq \max\{p, 2n^2 + n - 1\}$ (see Lemma 2 in the supplementary material). Here, the first term ($p$) represents the maximal rank of $\boldsymbol{\Lambda}$, and the second term ($2n^2 + n - 1$) is the maximal CP-rank of $\boldsymbol{P}_{\alpha\beta}\boldsymbol{\Lambda}\boldsymbol{P}_{\alpha\beta}^\top$.

Thus, for any $m \geq R$, we can reformulate (CP-NN) by expressing $\boldsymbol{\Lambda}$ as $\sum_{i=1}^m \boldsymbol{\lambda}_j \boldsymbol{\lambda}_j^\top$, with the constraint $\boldsymbol{P}_{\alpha\beta}\boldsymbol{\lambda}_j \geq \boldsymbol{0}_{2n\times 1}$, without changing the global solution:

$$\min_{\boldsymbol{\lambda}_j \in \mathbb{R}^p} \quad \left\| \sum_{j=1}^m \boldsymbol{P}_\alpha \boldsymbol{\lambda}_j \boldsymbol{\lambda}_j^\top \boldsymbol{P}_v^\top - \boldsymbol{Y} \right\|_F^2 + \frac{\gamma}{2} \sum_{j=1}^m \left( \|\boldsymbol{P}_u \boldsymbol{\lambda}_j\|_2^2 + \|\boldsymbol{P}_v \boldsymbol{\lambda}_j\|_2^2 \right), \tag{4}$$

$$\text{s.t.} \quad \boldsymbol{P}_{\alpha\beta}\boldsymbol{\lambda}_j \geq \boldsymbol{0}_{2n\times 1}, \quad \sum_{j=1}^m \left( \mathrm{trace}(\boldsymbol{P}_\alpha \boldsymbol{\lambda}_j \boldsymbol{\lambda}_j^\top \boldsymbol{P}_\beta^\top) + \|\boldsymbol{M}\boldsymbol{\lambda}_j\|_2^2 \right) = 0,$$

where we used the fact that $\mathrm{trace}(\boldsymbol{a}\boldsymbol{b}^\top) = \mathrm{trace}(\boldsymbol{b}^\top \boldsymbol{a})$, which further simplifies to $\|\boldsymbol{a}\|_2^2$ when $\boldsymbol{a} = \boldsymbol{b}$. We have omitted the $\mathcal{PSD}$ and $\mathcal{CP}$ constraints since they are inherently satisfied by the factorization.

Problems (3) and (4) share the same objective function. Next, we show that their feasible sets are also identical. Note that since each term in the summation constraint of problem (4) is non-negative, the sum can be zero only if each individual term is zero. We complete the proof by noting $\|\boldsymbol{M}\boldsymbol{\lambda}_j\|_2^2 = 0$ if and only if $\boldsymbol{M}\boldsymbol{\lambda}_j = \boldsymbol{0}_{n\times 1}$. For more details on the proof, we refer the reader to Appendix B. $\square$

**Remark 1.** *Although we focused on the MSE loss for simplicity, our results extend to any convex loss function $\ell : \mathbb{R}^d \times \mathbb{R}^c \to \mathbb{R}$.*

### 3.1 THE INFINITE WIDTH REGIME

**Definition 9** (Infinite-width Two Layer RELU Networks). *An infinite-width fully connected two-layer ReLU network can be expressed as a Lebesgue integral over a signed measure $\nu(\boldsymbol{u}, \boldsymbol{v})$ defined on the weights $\boldsymbol{u}$ and $\boldsymbol{v}$, subject to the condition that $(\boldsymbol{x}^\top \boldsymbol{u})_+ \boldsymbol{v}$ is integrable w.r.t. $\nu$ for any $\boldsymbol{x} \in \mathbb{R}^d$:*

$$\psi(\boldsymbol{x}) = \int_{\mathbb{R}^d \times \mathbb{R}^c} (\boldsymbol{x}^\top \boldsymbol{u})_+ \boldsymbol{v}^\top d\nu(\boldsymbol{u}, \boldsymbol{v}). \tag{NN$\int$}$$

A similar characterization of infinite-width neural networks can be found in (Mhaskar, 2004; Bach, 2017). For our analysis, we restrict $\nu(\boldsymbol{u}, \boldsymbol{v})$ to be a probability measure induced over the product space, $\nu : \mathcal{B}^{d+c} \to [0, 1]$, where $\mathcal{B}^n$ is the Borel $\sigma$-algebra over $\mathbb{R}^n$.

While the case of countably infinite number of hidden neurons, which is represented by an infinite sum in Eq. (NN), is captured as a special case of Eq. (NN$\int$) with a discrete probability measure, Eq. (NN$\int$) also accommodates other notions of infinite-width networks, as the underlying probability measure can be continuous or a combination of discrete and continuous components.

The training of an infinite-width ReLU neural network given in Eq. (NN$\int$) with the mean squared error loss can be expressed as follows:

$$\min_{\nu : \mathcal{B}^{d+c} \to [0,1]} \left\| \int_{\mathbb{R}^d \times \mathbb{R}^c} (\boldsymbol{X}\boldsymbol{u})_+ \boldsymbol{v} d\nu(\boldsymbol{u}, \boldsymbol{v}) - \boldsymbol{Y} \right\|_F^2 + \frac{\gamma}{2} \int_{\mathbb{R}^d \times \mathbb{R}^c} (\|\boldsymbol{u}\|_2^2 + \|\boldsymbol{v}\|_2^2) d\nu(\boldsymbol{u}, \boldsymbol{v}), \quad \text{(NN$\int$-Train)}$$

where optional Tikhonov regularization is controlled by the parameter $\gamma \geq 0$. The following theorem presents our second main result, which shows that the training problem (NN∫-Train) is also equivalent to our CP formulation.

**Theorem 2.** *Consider the two-layer infinite-width ReLU neural network defined in Eq. (NN∫). For any input dimension $d$ and output dimension $c$, the training problem (NN∫-Train) is equivalent to the convex optimization problem (CP-NN), implying that they share the same optimal values.*

We refer the reader to Appendix C for the proof. This result implies that the infinite-width ReLU network training problem (NN∫-Train) can be solved to global optimality by training a finite-width ReLU network with a width exceeding the critical threshold defined in Theorem 1. The resulting solution can then be represented as a discrete probability measure with finite support.

## 4 SDP RELAXATION FOR TRAINING TWO-LAYER ReLU NETWORKS

Despite its convexity, solving (CP-NN) remains intractable due to the complete positivity constraint. To tackle this challenge, we propose an SDP relaxation.

**Proposition 1** (SDP-NN). *The following SDP is a relaxation of the (CP-NN) problem, obtained by replacing the completely positive cone with the doubly non-negative cone, resulting in the constraints $\boldsymbol{P}_{\alpha\beta}\boldsymbol{\Lambda}\boldsymbol{P}_{\alpha\beta}^\top \in \mathcal{N}$ and $\boldsymbol{P}_{\alpha\beta}\boldsymbol{\Lambda}\boldsymbol{P}_{\alpha\beta}^\top \in \mathcal{PSD}$. The latter is omitted as it is inherently satisfied for all $\boldsymbol{\Lambda} \in \mathcal{PSD}$:*

$$\min_{\boldsymbol{\Lambda} \in \mathbb{R}^{p \times p}} \quad \left\| \boldsymbol{P}_\alpha \boldsymbol{\Lambda} \boldsymbol{P}_v^\top - \boldsymbol{Y} \right\|_F^2 + \frac{\gamma}{2} \left( \text{trace}(\boldsymbol{P}_u \boldsymbol{\Lambda} \boldsymbol{P}_u^\top) + \text{trace}(\boldsymbol{P}_v \boldsymbol{\Lambda} \boldsymbol{P}_v^\top) \right)$$

$$\text{s.t.} \quad \text{trace}(\boldsymbol{P}_\alpha \boldsymbol{\Lambda} \boldsymbol{P}_\beta^\top) + \text{trace}(\boldsymbol{M} \boldsymbol{\Lambda} \boldsymbol{M}^\top) = 0 \tag{SDP-NN}$$

$$\boldsymbol{\Lambda} \in \mathcal{PSD}$$

$$\boldsymbol{P}_{\alpha\beta} \boldsymbol{\Lambda} \boldsymbol{P}_{\alpha\beta}^\top \in \mathcal{N}.$$

**Remark 2.** *This is essentially the zeroth-order Sum-of-Squares (SoS) relaxation of (CP-NN). While tighter relaxations could be constructed using the SoS hierarchy (Parrilo, 2000), the computational requirements often make these higher-order formulations impractical.*

After solving (SDP-NN), a rounding procedure is required to extract the weights of the trained neural network from the lifted solution $\boldsymbol{\Lambda}_\star$.

**Proposition 2** (Rounding). *Selecting the rounding width $R$ as the critical width of the network, we can formulate the rounding problem as follows:*

$$\min_{\boldsymbol{\lambda} \in \mathbb{R}^{p \times R}} \quad \phi(\boldsymbol{\lambda}) := \left\| \boldsymbol{\Lambda}_\star - \boldsymbol{\lambda}\boldsymbol{\lambda}^\top \right\|_F^2 \tag{5}$$

$$\text{s.t.} \quad \boldsymbol{M}\boldsymbol{\lambda} = \boldsymbol{0}_{n \times R}, \quad \boldsymbol{P}_\alpha \boldsymbol{\lambda} \odot \boldsymbol{P}_\beta \boldsymbol{\lambda} = \boldsymbol{0}_{n \times R}, \quad \boldsymbol{P}_{\alpha\beta} \boldsymbol{\lambda} \geq \boldsymbol{0}_{2n \times R},$$

*where $\odot$ denotes the Hadamard product. The first and second layer weights can be recovered from the rounded solution as $\boldsymbol{P}_u \boldsymbol{\lambda}$ and $\boldsymbol{P}_v \boldsymbol{\lambda}$, respectively.*

It is straightforward to verify that the feasible set of problem (5) coincides with that of problem (3), which is, in turn, equivalent to the feasible set of the (CP-NN) problem. Consequently, this formulation aims to identify the closest point $\boldsymbol{\lambda}\boldsymbol{\lambda}^\top$ to the SDP solution $\boldsymbol{\Lambda}_\star$ within the feasible set of (CP-NN). While finding a global solution is intractable due to the non-convexity of the objective function and constraints, we propose an effective heuristic based on the three-operator splitting (TOS) method (Davis & Yin, 2017). We define the two sets comprising the constraints as:

$$\mathcal{D}_1 := \{ \boldsymbol{\lambda} \in \mathbb{R}^{p \times R} : \boldsymbol{P}_\alpha \boldsymbol{\lambda} \odot \boldsymbol{P}_\beta \boldsymbol{\lambda} = \boldsymbol{0}_{n \times R}, \ \boldsymbol{P}_{\alpha\beta} \boldsymbol{\lambda} \geq \boldsymbol{0}_{2n \times R} \}$$

$$\mathcal{D}_2 := \{ \boldsymbol{\lambda} \in \mathbb{R}^{p \times R} : \boldsymbol{M}\boldsymbol{\lambda} = \boldsymbol{0}_{n \times R} \}$$

Then, we can apply TOS for the rounding, starting from an initial estimate $\bar{\boldsymbol{\lambda}}^0 \in \mathbb{R}^{p \times R}$ and iteratively updating it by the following formula:

$$\boldsymbol{\lambda}^k = \text{proj}_{\mathcal{D}_1}(\bar{\boldsymbol{\lambda}}^k)$$

$$\hat{\boldsymbol{\lambda}}^k = \text{proj}_{\mathcal{D}_2}(2\boldsymbol{\lambda}^k - \bar{\boldsymbol{\lambda}}^k - \eta \nabla \phi(\boldsymbol{\lambda}^k)) \tag{TOS}$$

$$\bar{\boldsymbol{\lambda}}^{k+1} = \bar{\boldsymbol{\lambda}}^k - \boldsymbol{\lambda}^k + \hat{\boldsymbol{\lambda}}^k$$

Table 1: Summary of datasets, their sizes, and dimensions of the corresponding SDP relaxations.

| Dataset | # Instances | # Inp. Feat | # Out. Feat | # Variables | # Constraints |
|---------|-------------|-------------|-------------|-------------|---------------|
| Random | 25 | 2 | 5 | 3249 | 8999 |
| Spiral | 60 | 2 | 3 | 15625 | 45651 |
| Iris | 75 | 4 | 3 | 24649 | 71799 |
| Ionosphere | 175 | 34 | 2 | 148996 | 420493 |
| Pima Indians | 383 | 8 | 2 | 602176 | 1791109 |
| Bank Notes | 685 | 4 | 2 | 1893376 | 5663653 |
| MNIST | 1000 | 20 | 10 | 4120900 | 12743300 |

*Note: The # Instances column shows the number of training samples.*
*All real datasets are split into $50\%$ train and $50\%$ test sets.*

where $\eta > 0$ is the step-size parameter. We can compute the gradient by $\nabla \phi(\boldsymbol{\lambda}) = 4(\boldsymbol{\lambda}\boldsymbol{\lambda}^\top - \boldsymbol{\Lambda}_\star)\boldsymbol{\lambda}$. $\mathrm{proj}_{\mathcal{D}_2}(\boldsymbol{\lambda}) = (\boldsymbol{I} - \boldsymbol{M}^\dagger \boldsymbol{M})\boldsymbol{\lambda}$ projects $\boldsymbol{\lambda}$ onto the nullspace of $\boldsymbol{M}$. And $\mathrm{proj}_{\mathcal{D}_1}$ can be computed as

$$
\mathrm{proj}_{\mathcal{D}_1}(\boldsymbol{\lambda})_{i,j} = \begin{cases} 0 & 1 \leq i \leq 2n, \ \boldsymbol{\lambda}_{i,j} < 0 \\ 0 & 1 \leq i \leq n, \ \boldsymbol{\lambda}_{i+n,j} > \boldsymbol{\lambda}_{i,j} \\ 0 & n+1 \leq i \leq 2n, \ \boldsymbol{\lambda}_{i-n,j} > \boldsymbol{\lambda}_{i,j} \\ \boldsymbol{\lambda}_{i,j} & \text{otherwise} \end{cases} \tag{6}
$$

**Remark 3.** *We employ the rounding step primarily to demonstrate that the solution obtained from the SDP relaxation is reasonable. Our main objective is to assess the quality of the SDP relaxation, not to guarantee convergence in the rounding phase. To the best of our knowledge, there are no known convergence guarantees for TOS in our setting, as existing results for TOS in non-convex optimization apply only to smooth non-convex terms (Bian & Zhang, 2021; Yurtsever et al., 2021a) and do not extend to non-convex constraint sets. Nonetheless, TOS has proven to be an effective heuristic in our experiments, validating the (SDP-NN) relaxation. The ability to extract trained neural network weights confirms that the (SDP-NN) solution encodes the necessary information. Importantly, the rounding step is not integral to our theoretical contributions and is used only for empirical evaluation.*

## 5 NUMERICAL EXPERIMENTS

We conduct a series of experiments over synthetic and real datasets to investigate the empirical tightness of our SDP relaxation.

**Datasets**. We use the following datasets in our experiments:

*Random*: A synthetic dataset with a data matrix $\boldsymbol{X} \in \mathbb{R}^{25 \times 2}$ and labels $\boldsymbol{Y} \in \mathbb{R}^{25 \times 5}$, generated using a 2-layer neural network with 100 hidden neurons. We create 100 random datasets by initializing the entries of $\boldsymbol{X}$ and the generator network's weights using a standard normal distribution.

*Spiral*: An artificially-generated 3-class classification dataset, described by (Sahiner et al., 2021). It includes 60 samples (20 per class), with a data matrix $\boldsymbol{X} \in \mathbb{R}^{60 \times 2}$ containing 2 input features and one-hot encoded labels $\boldsymbol{Y} \in \mathbb{R}^{60 \times 3}$.

*Iris (Fisher, 1936)*: The dataset consists of 150 samples of iris flowers from three different classes, each sample is described by four features. The training partition includes a data matrix $\boldsymbol{X} \in \mathbb{R}^{75 \times 4}$ and one-hot encoded labels $\boldsymbol{Y} \in \mathbb{R}^{75 \times 3}$.

*Ionosphere (Sigillito et al., 1989)*: A radar dataset with 351 instances and 34 input features for binary classification, with class imbalance. The training partition includes a data matrix $\boldsymbol{X} \in \mathbb{R}^{175 \times 34}$ and one-hot encoded labels $\boldsymbol{Y} \in \mathbb{R}^{175 \times 2}$.

*Pima Indians Diabetes (Smith et al., 1988)*: A diabetes prediction dataset that consists of 768 patients, with 8 medical predictors as features, and a binary classification task, with class imbalance. Out of 768 instances, we have 268 positive instances. The training partition includes a data matrix $\boldsymbol{X} \in \mathbb{R}^{383 \times 8}$ and one-hot encoded labels $\boldsymbol{Y} \in \mathbb{R}^{383 \times 2}$.

Table 2: SGD Loss, Approximation Ratio (AR) (%), and Runtime (sec) of SDP-NN.

| Dataset | $\gamma$ | Training Objective | | | | | | AR | Runtime |
|---|---|---|---|---|---|---|---|---|---|
| | | SGD-5 | SGD-10 | SGD-100 | SGD-200 | SGD-300 | SDP-NN | SDP-NN | SDP-NN |
| Random | 0.1 | 18.27 ±5.76 | 8.60 ±1.16 | 8.09 ±1.04 | 8.09 ±1.04 | 8.09 ±1.04 | 7.28 ±0.98 | 89.93 | 11.46 |
| | 0.01 | 11.78 ±5.53 | 1.32 ±0.25 | 0.94 ±0.12 | 0.94 ±0.12 | 0.94 ±0.12 | 0.76 ±0.10 | 80.66 | 14.85 |
| Spiral | 0.1 | 16.64 | 16.59 | 16.59 | 16.59 | 16.59 | 16.24 | 97.84 | 1566.65 |
| | 0.01 | 15.56 | 15.21 | 15.16 | 15.16 | 15.16 | 11.66 | 76.90 | 923.83 |

*Bank Notes Authentication (Lohweg, 2012)*: A binary classification dataset with 1372 instances, where features extracted using wavelet transforms are used to determine whether a banknote is genuine or forged. The training data includes $X \in \mathbb{R}^{685 \times 4}$ and one-hot encoded labels $Y \in \mathbb{R}^{685 \times 2}$.

*MNIST (LeCun et al., 2010)*: A down-sampled and dimensionally-reduced version of the popular image classification dataset, consisting of 1000 instances with 20 features each obtained using Principal Component Analysis (PCA). The training data includes $X \in \mathbb{R}^{1000 \times 20}$ and one-hot encoded labels $Y \in \mathbb{R}^{1000 \times 10}$.

Table 1 provides an overview of the dataset sizes and the dimensionality of the corresponding problem size of (SDP-NN). All real datasets were split 50-50 into train and test partitions.

**Computational environment & complexity**. The experiments were conducted on a Intel Xeon Gold 6132 with 192 GB of RAM and 2x14 cores. Solving CP-NN or the associated rounding problems is NP-hard. The complexity of solving an SDP depends on the specific algorithm used. Our SDP formulations were solved by CVXPY (Diamond & Boyd, 2016), employing either the interior point method (IPM) solver by MOSEK (Andersen & Andersen, 2000), or the Splitting Cone Solver (SCS) (O'donoghue et al., 2016), depending on the problem size. A typical interior point method for solving an SDP problem with an $n \times n$ matrix variable and $m$ constraints requires approximately $\mathcal{O}(n^3 + m^2 n^2 + m^3)$ arithmetic operations per iteration and about $\mathcal{O}(\log(1/\epsilon))$ iterations to achieve an $\epsilon$-accurate solution. While the formulation scales with $n^2$ (since the rank is absorbed), storing a CP factorization of the decision variable (i.e., the neural network weights) requires $\mathcal{O}(n^3)$ storage.

## 5.1 EMPIRICAL APPROXIMATION RATIO OF THE SDP RELAXATION

We evaluate the tightness of our SDP relaxation by comparing its optimal objective value to the training loss obtained using Stochastic Gradient Descent (SGD) on (NN-Train). Due to the difficulty of approximating the global optimum with SGD on real datasets, we conduct this experiment using the synthetic *Random* and *Spiral* datasets.

The results are summarized in Table 2. For each dataset, we solve the training problem using regularization parameters $\gamma = 0.1$ and $\gamma = 0.01$. The table columns show the training loss achieved by SGD for various hidden layer sizes, ranging from $m = 5$ to $m = 300$, along with the training loss achieved by the SDP-NN formulation. For the *Random* dataset, the results are averaged over 100 trials, with error bars indicating standard deviations.

The last column reports the **approximation ratio** (AR) of the SDP relaxation, computed as the ratio of the loss obtained by SDP-NN to the loss achieved by SGD with 300 hidden neurons. Empirically, we find that the approximation ratio is above $76.90\%$ in these experiments. We also observe that the ratio improves for higher regularization parameters, which is expected, as stronger regularization tends to smooth the optimization landscape and reduces the impact of non-convexity.

**Remark 4.** *Ratio reported in Table 2 represents only a lower bound on the actual approximation ratio; since, (i) SGD may converge to a local solution, and (ii) 300 hidden neurons used in this experiment is smaller than the actual critical width, which scales quadratically with the number of data points, and quickly exceeds practical limits.*

To partially address these challenges, we ran SGD five times with different random initializations for each configuration and selected the best outcome. Even so, there is no guarantee of reaching a global optimum. Similarly, although $m = 300$ is below the critical width, we observe that the results tend to saturate, and increasing $m$ further does not lead to any significant improvements for SGD. Despite these limitations, SGD training can provide an upper bound on the optimal value for infinite-width neural network training. To complement this, we compute a lower bound on the optimal values of

Table 3: Evaluating prediction quality (F1 Score & Accuracy) and runtime (seconds).

| Method | Iris | | Ionosphere | | Pima Indians | | Bank Notes | | MNIST | |
|---|---|---|---|---|---|---|---|---|---|---|
| | $\gamma = 0.1$ | $\gamma = 0.01$ | $\gamma = 0.1$ | $\gamma = 0.01$ | $\gamma = 0.1$ | $\gamma = 0.01$ | $\gamma = 0.1$ | $\gamma = 0.01$ | $\gamma = 0.1$ | $\gamma = 0.01$ |
| **Weighted Average F1 Score** | | | | | | | | | | |
| SGD | 0.96 | 0.987 | 0.915 | 0.898 | 0.626 | 0.594 | **0.993** | 0.992 | 0.880 | 0.863 |
| NNGP | 0.866 | 0.975 | 0.919 | **0.928** | 0.690 | 0.683 | 0.978 | **0.993** | 0.903 | **0.919** |
| NTK | **0.993** | 0.993 | **0.924** | 0.920 | 0.668 | 0.672 | 0.991 | **0.993** | **0.907** | 0.915 |
| SDP-NN | 0.987 | 0.987 | **0.924** | 0.927 | 0.679 | 0.703 | 0.930 | 0.893 | 0.862 | 0.849 |
| SDP-NN-bias | 0.946 | **1.000** | 0.912 | 0.921 | **0.714** | **0.744** | 0.991 | 0.985 | 0.858 | 0.838 |
| **Overall Test Accuracy** | | | | | | | | | | |
| SGD | 0.960 | 0.987 | 0.891 | 0.88 | 0.583 | 0.557 | 0.988 | **0.985** | 0.818 | 0.796 |
| NNGP | 0.827 | 0.973 | 0.886 | 0.886 | 0.534 | 0.602 | 0.966 | 0.983 | 0.858 | **0.879** |
| NTK | **0.987** | 0.987 | 0.886 | 0.897 | 0.586 | 0.620 | 0.983 | **0.985** | **0.863** | 0.876 |
| SDP-NN | **0.987** | 0.987 | **0.920** | 0.909 | 0.646 | 0.625 | 0.860 | 0.767 | 0.794 | 0.778 |
| SDP-NN-bias | 0.947 | **1.000** | 0.909 | **0.914** | **0.672** | **0.703** | **0.991** | 0.980 | 0.791 | 0.76 |
| **Runtime (sec)** | | | | | | | | | | |
| SDP-NN | 39 | 171 | 4416 | 8150 | 12256 | 13564 | 102919 | 101952 | 66157 | 155810 |

the SDP formulations using the dual objectives in CVXPY, utilizing the interior-point solver from MOSEK. Together, these bounds allow us to establish a reliable approximation lower-bound for the relaxation.

The training loss curves of SGD for each experiment, along with implementation details such as learning rate and number of epochs, are provided in Appendix E.

## 5.2 PREDICTION PERFORMANCE OF THE SDP RELAXATION

We assess the prediction quality of our SDP relaxation combined with the TOS-based rounding procedure on real datasets using regularization parameters $\gamma = 0.1$ and $\gamma = 0.01$. Due to computational constraints, we set the rank $R$ in the rounding procedure to 300, which is lower than the actual critical width. The rounding procedure is initialized using the square root of the SDP-NN solution, obtained via SVD decomposition. We then adjust the resulting factor matrices by either truncating or expanding them to dimensions $\mathbb{R}^{p \times 300}$. The TOS step size is set as $\eta = 1/\|\mathbf{\Lambda}_\star\|_2$, where $\mathbf{\Lambda}_\star$ denotes the solution to the SDP-NN, and the algorithm is run for 1000 iterations.

We benchmark our test accuracy against the NNGP and NTK kernel methods and include results from SGD with 300 hidden neurons as a baseline. The results of this experiment are summarized in Table 3, where the weighted average F1 score and test accuracy are used as the performance metrics. For classification, the threshold was varied from 0 to 1 in increments of 0.01, and the threshold that maximized the overall test accuracy was selected. Overall, the predictions obtained from the SDP relaxation are competitive across most settings.

**Effect of bias.** We also experimented with incorporating a bias term to the first layer. As expected, Table 3 (SDP-NN-bias) shows a general increase in terms of the weighted average F1 score and overall test accuracy.

## 6 RELATED WORKS

We now review the literature on (in)finite width neural networks as well as convex optimization techniques for training them.

**Convex optimization for neural network training**. Early discussions of convex neural nets can be found in (Bengio et al., 2005; Bach, 2017; Fang et al., 2022). These initial studies primarily focus on networks with infinite width, which leads to optimization problems in an infinite-dimensional space.

Sahiner et al. (2021) developed convex equivalents for the non-convex NN training problem with ReLU activations, based on a theory of convex semi-infinite duality. Different than ours, their approach involves an explicit summation over all possible sign patterns in the ReLU layer. Since the number of distinct sign patterns grows exponentially with the rank of the data matrix, their analysis is primarily focused on data matrices of fixed small rank or the *spike-free* structures. Building on

this duality theory, Ergen & Pilanci (2021a) derived convex optimization formulations for two- or three-layer convolutional neural networks (CNNs) with ReLU activations. Bartan & Pilanci (2021a) explored semidefinite lifting for training neural networks with polynomial activations, which is further extended to quantized neural networks with polynomial activations in (Bartan & Pilanci, 2021b), and to deep neural networks with polynomial and ReLU activations in (Bartan & Pilanci, 2023). Ergen & Pilanci (2021c) demonstrated that training multiple three-layer ReLU regularized sub-networks can be equivalently cast as a convex optimization problem in a higher-dimensional space. Ergen & Pilanci (2021b;d) showed that optimal hidden-layer weights for two-layer and deep ReLU neural networks are the extreme points of a convex set. They further applied these methods in transfer learning tasks with large language models. Sahiner et al. (2022) developed convex optimization problems for vision transformers, focusing on a single self-attention block with ReLU activation. Finally, Wang et al. (2024) introduced randomized algorithms from a geometric algebra perspective to efficiently address the enumeration of sign patterns. Sahiner et al. (2024) applied the Burer-Monteiro factorization (Burer & Monteiro, 2003) to efficiently solve these formulations at a larger scale.

Our mathematical techniques differ from prior work. Unlike (Sahiner et al., 2021), we do not require enumeration or sampling of sign patterns, nor do we rely on the aforementioned duality theory. Unlike (Fang et al., 2022), we do not assume activation functions with continuously differentiable gradients, and formulate the training of finite- and infinite-width ReLU neural networks as a finite-dimensional convex optimization problem, assuming fixed input/output dimensions and a set number of data points, characterizing a finite critical width. Since our CP formulation is exact yet intractable, we propose SDP relaxations that can be solved in polynomial time.

**Infinite-Width Neural Networks (IWNNs)**. Investigations of IWNNs address fundamental questions about the limits of their learning ability and the types of functions they can approximate. This focus is also relevant as overparameterized networks, which are commonly used in practice, often demonstrate strong generalization to unseen data (Zhang et al., 2021). Another motivation for studying IWNNs is their connection to other areas of machine learning, such as Gaussian processes (Neal, 2012). There are two main approaches for training infinite-width fully connected neural networks in the literature: weakly-trained networks and fully-trained networks.

Weakly-trained networks are those where only the top classification layer is trained, while all other layers remain randomly initialized. Lee et al. (2018); Matthews et al. (2018) studied fully-connected weakly-trained IWNNs, while Novak et al. (2019); Garriga-Alonso et al. (2019) focused on convolutional variants. Yang (2019) extended this analysis to other weakly-trained infinite-width architectures. All these works adopt a Gaussian process (GP) perspective of weakly-trained IWNNs.

In contrast, fully-trained networks are those in which all the layers are trained, using variations of gradient descent. Jacot et al. (2018) introduced Neural Tangent Kernel (NTK), which captures the behavior of fully-connected deep neural networks in the infinite-width limit, trained using gradient descent. The NTK is different from the GP kernels: it is defined by the gradient of a fully connected network's output with respect to its weights, based on random initialization. Arora et al. (2019) gave a rigorous, non-asymptotic proof that the NTK captures the behavior of a fully-trained wide neural network under weaker conditions and introduced the concept of convolutional neural tangent kernel (CNTK). They also developed an exact and efficient dynamic programming algorithm to compute CNTKs for ReLU activation. Lee et al. (2020) conducted an empirical study comparing the Neural Network Gaussian Process (NNGP) kernel (Lee et al., 2018) and the NTK in both finite-width and IWNNs. Unlike these, we do not adopt a GP-based view or kernel methods. Instead, we model fully-trained fully-connected IWNNs using a CP formulation and corresponding computationally tractable SDP relaxations.

Another line of work focuses on the learning theory of infinite-width neural networks, including universal approximation bounds (e.g., (Barron, 1993; Mhaskar, 2004; Klusowski & Barron, 2018)), representer theorems (e.g., (Parhi & Nowak, 2021; Bartolucci et al., 2023; Shenouda et al., 2024)), and capacity control results (e.g., (Ongie et al., 2019; Savarese et al., 2019)). Of particular relevance, in their recent work, Shenouda et al. (2024) showed that an infinite dimensional learning problem in variation spaces (a special class of reproducing kernel Banach spaces) can be solved by training a finite-width neural network with width greater than the square of the number of training data points. Intriguingly, this result draws parallels with our critical width $R$, which also scales quadratically with the number of training data points, although the underlying assumptions and analytical techniques in the two works are significantly different.

**Other related works**. We address the non-convex problem of training two-layer ReLU neural networks by transforming it into a convex optimization problem in a higher-dimensional space. This approach belongs to a class of techniques known as *lifting*, where the decision variable $w$ is replaced with a quadratic term $W = ww^\top$, enabling the convex reformulation or relaxation of an otherwise non-convex problem, see (Bomze et al., 2000; Burer, 2009; Bao et al., 2011; Anstreicher, 2012) and references therein. In the context of neural networks, Brown et al. (2022) applied lifting for the verification of neural network outputs. A key distinction between our formulation and the prior work in (Burer, 2009; Brown et al., 2022) lies in the structure of the objective function. In previous work, the objective function is quadratic in the original space and simplifies to a linear objective in the lifted space, which plays a crucial role in the analysis, as solutions to linear minimization problems always appear at extreme points of the feasible set. In our problem, the objective function is convex in the lifted space (rather than linear), and we establish an exact correspondence between the feasible sets in the original and lifted spaces for both wide and infinite-width networks.

Our work should not be confused with the line of research introduced by Askari et al. (2018) known as *lifted neural networks*, as the notion of lifting in our work differs from theirs. Both the goals and approaches are distinct. They reformulate the problem as a biconvex optimization, enabling layer-wise training via block coordinate descent. They do not introduce the quadratic terms that are central to our approach, and their resulting problem formulation remains non-convex.

## 7 CONCLUSIONS AND FUTURE DIRECTIONS

We derived a convex optimization formulation for the training problem of a two-layer ReLU neural network with a sufficiently wide hidden layer. Despite its convexity, the problem is intractable using classical computational methods due to the completely positive cone constraints. To address this, we proposed an SDP relaxation that can be solved in polynomial time using off-the-shelf SDP solvers, combined with a rounding heuristic. Notably, the size of our formulation is independent of the network width. While previous work has established the critical width for universal approximation of continuous functions with neural networks (Cai, 2022), we established limits on the expressivity of wide-width neural networks in terms of the data size as our critical width $R$ is based on the size of the data. These type of findings can help advance theoretical understanding of neural networks.

**Limitations**. An obvious limitation of the proposed framework is its lack of scalability, a common issue for many SDP formulations in machine learning applications. Since the problem size scales quadratically with the number of data points, solving these formulations at the scale required for real-world neural network applications poses a significant computational challenge. Nevertheless, our findings offer valuable theoretical insights into ReLU networks, for instance, allowing us to interpret the network width as the (CP-)rank of the corresponding completely positive program.

**Future work**. We proposed an exact CP formulation, which is computationally intractable for classical computers. However, rapid progress in quantum computing poses a promising alternative (Birdal et al., 2021a). In particular, hybrid classical-quantum algorithms have demonstrated potential for solving specific classes of copositive programs (Yurtsever et al., 2022; Brown et al., 2024) as well as training binary versions of neural networks (Krahn et al., 2024). Developing and implementing algorithms—even as proof-of-concept experiments—to train general ReLU networks on these platforms represents a compelling research direction.

Over-parameterized networks are often easier to optimize using traditional methods such as SGD (Livni et al., 2014). Although it is possible to construct compact networks with fewer parameters that generalize comparably, compressed models are significantly more challenging to optimize in the reduced-dimensional space (Arora et al., 2018). A similar phenomenon is observed in the SDP literature: Waldspurger & Waters (2020) showed that even when an SDP has a unique rank-1 solution, factorizing the variable with a rank below a certain threshold may introduce spurious local minima. Given these parallels, a natural question arises: is there a corresponding bound for completely positive programs, and could this provide insights into the required width of ReLU networks?

## ETHICS & REPRODUCIBILITY

**Ethics**. This work adheres to the ICLR Code of Ethics, ensuring responsible use of publicly available datasets and maintaining full transparency in experimental design and reporting. The study complies with ethical guidelines on research integrity and considers potential societal impacts. Since the contributions are primarily theoretical, the focus is on advancing our understanding of the theory of artificial intelligence rather than targeting specific applications that may have direct societal impacts.

**Reproducibility statement**. All datasets used in this work are referenced and publicly accessible. The experimental configurations and computational environment are outlined in detail within the main text and the appendix. We make our implementation available under https://github.com/KarthikPrakhya/SDPNN-IW. Furthermore, all random seeds are provided to facilitate precise reproducibility.

## ACKNOWLEDGMENTS

AY and KP were supported by the Wallenberg AI, Autonomous Systems and Software Program (WASP) funded by the Knut and Alice Wallenberg Foundation. We thank the High Performance Computing Center North (HPC2N) at Umeå University for providing computational resources and valuable support during test and performance runs. The computations were enabled by resources provided by the National Academic Infrastructure for Supercomputing in Sweden (NAISS), partially funded by the Swedish Research Council through grant agreement no. 2022-06725. TB was supported by a UKRI Future Leaders Fellowship [grant number MR/Y018818/1]. We acknowledge the use of OpenAI's ChatGPT for editorial assistance in preparing this manuscript.

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

APPENDIX

We will first provide additional discussions before moving onto the proofs and additional experiments.

## A   DISCUSSIONS

**On the contributions to theoretical understanding of neural networks**. Convex formulations of neural networks can offer significant benefits in advancing our understanding of deep networks. We now provide a brief exposition and perspective on this:

- **Interpretability.** Convex formulations are essential in converting pesky non-convex landscapes into interpretable ones. By doing so, they provide a clearer view of the solution space, revealing its geometric structure, and help identify properties of optimal solutions without getting trapped in suboptimal configurations.
- **Rigorous analysis.** They pave the way for a rigorous theoretical analysis, including guarantees of convergence, uniqueness, and stability of solutions. This is in contrast to traditional training where guarantees are often limited or conditional on specific initialization schemes or overparameterization. We leave such analysis for a future work.
- **Lifting.** Thanks to our exact co-positive formulation, studying how high-dimensional lifting of data affects separability and decision boundaries can offer insights into how neural networks learn representations and resolve ambiguities in the data.
- **A holistic learning theory.** Bridging the gaps between optimization, geometry, and learning theory is fundamental to develop a modern theory for deep networks. This is what we precisely contribute towards with our theoretical formulation.

In the light of these, one particularly compelling avenue for future work involves **leveraging our convex formulation to study generalization**. Let us elaborate on this. Characterizing the loss landscape of neural networks is crucial to understanding the capacity and limitations of deep learning. Generalizing solutions are known to reside in the wide minima of the loss landscape (Hochreiter & Schmidhuber, 1997). This gives a relationship between the generalization error and the location of critical points. For our lifted convex formulations, the critical points are readily identifiable and can be mapped back to the non-convex landscape of the original problem through smooth parameterizations, as demonstrated by (Levin et al., 2024). By characterizing the local convexity of these points, we can potentially determine which solutions generalize better. While we have not yet investigated this, we plan to make such connections clearer in the future, where we further like to investigate the link between our formulation and the training dynamics (Andreeva et al., 2024; Birdal et al., 2021b).

On another frontier, the exact CP-NN formulation, combined with an analog of the results by Waldspurger and Waters on the tightness of the Pataki-Barvinok (PB) bound for factorization rank in CP-NN formulations, could provide insights into the minimal width required for ReLU neural networks to succeed (Waldspurger & Waters, 2020; Endor & Waldspurger, 2024). Specifically, these works studied the MaxCut SDP and its factorized (non-convex) formulation. It is well-established that the factorized (non-convex) problem does not exhibit spurious local minima when the factorization rank exceeds the PB bound, which scales as $\mathcal{O}(\sqrt{n})$, where $n$ is the number of vertices (number of constraints, for a more general SDP template). Recently, same authors demonstrated that the PB bound is tight, meaning that even if the original SDP problem has a unique rank-1 solution, the factorized (non-convex) problem can fail unless the factorization rank is greater than the PB bound. This result provides critical **guidance for selecting the rank in matrix factorization problems**.

**On NTK, NNGP and SGD.** NNGP models the prior predictive distribution of an untrained neural network in the infinite-width limit, while NTK captures the training dynamics of gradient descent under similar assumptions. Our formulation differs significantly from NTK and NNGP, and does not share the same constraints. Unlike NTK and NNGP, our convex formulation is exact and applies to both finite and infinite-width regimes as long as the critical-width threshold is satisfied. Additionally, our method is not dependent on initialization. Any empirical inexactness arises solely from the relaxation and rounding processes, which do not undermine the theoretical validity of what we present, even if the computation is NP-hard.

Neither NTK, NNGP or our work is equivalent to SGD. While NTK can approximate SGD dynamics under specific overparameterized conditions, this correspondence is not universal and relies on several assumptions. Importantly, as stated in the main paper, we use SGD with a hidden layer of 300 neurons in our experiments. Under these conditions, we do not expect SGD to align with NTK.

**On the losses and the lower bounds.** SDP-NN is a relaxation of the original problem, meaning it provides a lower bound on the objective. Therefore, its loss value is not directly comparable to SGD, which provides an upper bound by finding a feasible (but possibly suboptimal) solution. The lower loss for SDP-NN does not mean it *performs better* in the traditional sense, but rather indicates the quality of the relaxation. To evaluate the relationship between the two, we computed the approximation ratio (AR) in the main paper, quantifying how close the solution obtained by SGD is to the lower bound provided by SDP-NN. It is important to see that this is not a limitation of our method but stems from the nature of the problem – the global solution of the training being unavailable.

Let us elaborate on that a little. Specifically, the exact approximation ratio is defined as: $\mathrm{AR}_{\mathrm{exact}} = F^*_{\mathrm{relaxed}}/F^*$, where $F^*_{\mathrm{relaxed}}$ denotes the relaxed solution. However, since we do not know the true global minimum $F^*$, we instead compute the empirical AR, which provides only a lower bound on how well the SDP-NN relaxation captures the training problem: $\mathrm{AR} = F^*_{\mathrm{relaxed}}/F^*_{\mathrm{SGD}}$. Here, $F^*_{\mathrm{SGD}}$ corresponds to the local minimum found by SGD, and because $F^*_{\mathrm{SGD}} \geq F^*$, we ensure that $\mathrm{AR} \leq \mathrm{AR}_{\mathrm{exact}}$. This guarantees that the reported AR provides a lower bound on the actual approximation ratio.

**On rounding.** Although our rounding step lacks convergence guarantees, as noted in Remark 3, it serves as a valuable tool for validating our SDP-NN relaxation. Despite the heuristic nature of rounding, the ability to extract the trained weights of a successful neural network demonstrates that the SDP-NN solution contains the necessary information. It is important to emphasize that the rounding step is not integral to the theoretical contributions of the proposed framework and is used solely for empirical evaluations. It is possible to leverage other rounding mechanisms and we leave it as a future study to determine the optimal choice.

## B  PROOF OF THEOREM 1

Our formulation in Theorem 1 involves a $(p \times p)$ matrix decision variable $\boldsymbol{\Lambda} \in \mathcal{PSD}$, which contains a $(q \times q)$ principal submatrix $\boldsymbol{P}\boldsymbol{\Lambda}\boldsymbol{P}^\top \in \mathcal{CP}$, where $\boldsymbol{P} = \begin{bmatrix} \boldsymbol{I}_{q \times q} & \boldsymbol{0}_{q \times (p-q)} \end{bmatrix}$ is the row-selection operator. The following lemma plays a crucial role in our analysis, in showing that there exists a finite critical width for a two-layer ReLU network, beyond which increasing the number of hidden neurons does not improve the network's expressive power.

**Lemma 2.** *Let $\boldsymbol{\Lambda} \in \mathbb{R}^{p \times p}$ be a positive semidefinite matrix of rank $k$, with a completely positive principal submatrix $\widehat{\boldsymbol{\Lambda}} \in \mathbb{R}^{q \times q}$ of CP-rank $K$. Without loss of generality, we assume $\widehat{\boldsymbol{\Lambda}}$ is the leading principal submatrix of $\boldsymbol{\Lambda}$ given by $\widehat{\boldsymbol{\Lambda}} = \boldsymbol{P}\boldsymbol{\Lambda}\boldsymbol{P}^\top$ with $\boldsymbol{P} = \begin{bmatrix} \boldsymbol{I}_{q \times q} & \boldsymbol{0}_{q \times (p-q)} \end{bmatrix}$. Then, $\boldsymbol{\Lambda}$ can be decomposed as*

$$\boldsymbol{\Lambda} = \sum_{j=1}^{R} \boldsymbol{\lambda}_j \boldsymbol{\lambda}_j^\top, \quad \text{such that} \quad \boldsymbol{\lambda}_j \in \mathbb{R}^p \text{ and } \boldsymbol{P}\boldsymbol{\lambda}_j \in \mathbb{R}^q_+ \tag{7}$$

*for some $R \leq \max\{k, K\}$. As an immediate corollary, for any $(p \times p)$ PSD matrix with a $(q \times q)$ dimensional CP submatrix (without any restriction on their rank or CP-rank), we have $R \leq \max\{p, R_q\}$, where $R_q$ is the maximal CP-rank $R_q := q(q+1)/2 - 1$.*

We will use the following result (see Lemma 1 in (Xu, 2004)) in the proof of Lemma 2.

**Fact 1.** *If a PSD matrix $\boldsymbol{A} \in \mathbb{R}^{n \times n}$ admits two distinct factorizations, $\boldsymbol{A} = \boldsymbol{C}\boldsymbol{C}^\top$ and $\boldsymbol{A} = \boldsymbol{D}\boldsymbol{D}^\top$, where $\boldsymbol{C}, \boldsymbol{D} \in \mathbb{R}^{n \times N}$, then we can find an orthogonal matrix $\boldsymbol{Q} \in \mathbb{R}^{N \times N}$ such that $\boldsymbol{D} = \boldsymbol{C}\boldsymbol{Q}$.*

*Proof of Fact 1.* Let $\boldsymbol{c}_i \in \mathbb{R}^N$ and $\boldsymbol{d}_i \in \mathbb{R}^N$ be the $i$th rows of $\boldsymbol{C}$ and $\boldsymbol{D}$, respectively. Since $\{\boldsymbol{c}_i\}_{i=1}^n$ and $\{\boldsymbol{d}_i\}_{i=1}^n$ span the same subspace, there exists a linear transformation $\boldsymbol{Q}$ mapping $\mathrm{span}(\boldsymbol{c}_1, \ldots, \boldsymbol{c}_n)$ to $\mathrm{span}(\boldsymbol{d}_1, \ldots, \boldsymbol{d}_n)$ such that $\boldsymbol{d}_i = \boldsymbol{c}_i \boldsymbol{Q}$ for each $i = 1, \ldots, n$. By construction, we have $\langle \boldsymbol{d}_i, \boldsymbol{d}_j \rangle = \langle \boldsymbol{c}_i \boldsymbol{Q}, \boldsymbol{c}_j \boldsymbol{Q} \rangle = \langle \boldsymbol{c}_i, \boldsymbol{c}_j \rangle$, so $\boldsymbol{Q}$ preserves inner products on this subspace. Therefore, by extending $\boldsymbol{Q}$ to all of $\mathbb{R}^N$, we obtain an orthogonal matrix. $\square$

*Proof of Lemma 2.* For the PSD matrix $\boldsymbol{\Lambda}$ and any non-negative integer $R \geq k$, we have a factorization of the form $\boldsymbol{\Lambda} = \boldsymbol{B}\boldsymbol{B}^\top$, where $\boldsymbol{B} \in \mathbb{R}^{p \times R}$. This factorization is not necessarily unique, and the columns of $\boldsymbol{B}$ are not required to be linearly independent. Similarly, for the CP matrix $\widehat{\boldsymbol{\Lambda}}$, we have a factorization $\widehat{\boldsymbol{\Lambda}} = \boldsymbol{D}\boldsymbol{D}^\top$, where $\boldsymbol{D} \in \mathbb{R}^{q \times R}_+$ for any $R \geq K$. As before, this factorization is not necessarily unique, and the columns of $\boldsymbol{D}$ need not be linearly independent.

Consider these factorizations with $R = \max\{k, K\}$. Define $C$ as the first $q$ rows of $B$, given by $C = PB$. We have $\widehat{\Lambda} = CC^\top$. Then, by Fact 1, since any CP matrix is also PSD, there exists an orthogonal matrix $Q \in \mathbb{R}^{R \times R}$ such that $D = CQ$. Finally, let us define $E = BQ$. It is easy to verify that $EE^\top = \Lambda$, and $PE = D$ is entrywise non-negative. Hence, if we set $w_j$ equal to the $j$th column of $E$, we obtain the desired decomposition. $\square$

The following lemma will be used to handle the non-convexity introduced by ReLU activation:

**Lemma 3.** *For real-valued vector $\hat{\alpha} \in \mathbb{R}^n$ and non-negative real-valued vectors $\alpha, \beta \in \mathbb{R}^n_+$ with $\alpha := (\hat{\alpha})_+$ and $\beta := \alpha - \hat{\alpha}$, the following conditions are equivalent:*

$$\alpha = (\hat{\alpha})_+ \quad \text{if and only if} \quad \text{trace}(\alpha \beta^\top) = 0. \tag{8}$$

*Proof of Lemma 3.* First, observe that $\text{trace}(\alpha \beta^\top) = \text{trace}(\beta^\top \alpha) = \beta^\top \alpha = \sum_{i=1}^n \alpha_i \beta_i$.

Assume that the left-hand side in Eq. (8) holds. Note that if $\alpha$ represents the positive part of $\hat{\alpha}$, then $\beta$ is the negative part. Consequently, for each index $i$, either $\alpha_i$ or $\beta_i$ is zero, hence $\text{trace}(\alpha \beta^\top) = 0$.

Now, suppose that the right-hand side in Eq. (8) holds. Given that the elements of $\alpha$ and $\beta$ are non-negative, we have $\text{trace}(\alpha \beta^\top) = 0$ if and only if $\alpha_i \beta_i = \alpha_i(\alpha_i - \hat{\alpha}_i) = 0$ for all $i = 1, \dots, n$. This implies that if $\alpha_i > 0$, then $\hat{\alpha}_i = \alpha_i$. Otherwise, if $\alpha_i = 0$, we have $\hat{\alpha}_i = \alpha_i - \beta_i \leq 0$. Thus, we conclude that $\alpha = (\hat{\alpha})_+$. $\square$

*Proof of Theorem 1.* We start by rewriting (NN-Train) in the constrained form as

$$\min_{\substack{u_j \in \mathbb{R}^d, v_j \in \mathbb{R}^c \\ \alpha_j \in \mathbb{R}^n, \hat{\alpha}_j \in \mathbb{R}^n}} \left\| \sum_{j=1}^m \alpha_j v_j^\top - Y \right\|_F^2 + \frac{\gamma}{2} \sum_{j=1}^m (\|u_j\|_2^2 + \|v_j\|_2^2) \tag{9}$$

$$\text{s.t.} \quad \hat{\alpha}_j = X u_j \quad \text{and} \quad \alpha_j = (\hat{\alpha}_j)_+, \quad \text{for} \quad j = 1, \dots, m.$$

We introduce non-negative variables $\beta_j := \alpha_j - \hat{\alpha}_j$. As both $\alpha_j$ and $\beta_j$ are non-negative valued, it can be shown that $\alpha_j = (\hat{\alpha}_j)_+$ if and only if $\text{trace}(\alpha_j \beta_j^\top) = 0$, see Lemma 3. As a result, the constraints in Eq. (9) can be equivalently rewritten in terms of $\alpha_j$ and $\beta_j$ as:

$$\alpha_j, \beta_j \in \mathbb{R}^n_+, \quad \text{trace}(\alpha_j \beta_j^\top) = 0, \quad \text{and} \quad X u_j - \alpha_j + \beta_j = 0_{n \times 1}, \quad \text{for} \quad j = 1, \dots, m. \tag{10}$$

The quadratic terms in the objective function in Eq. (9) and the trace constraint in Eq. (10) introduce challenges due to non-convexity. We address this by lifting the problem into a higher-dimensional space, introducing new variables $\lambda_j \in \mathbb{R}^p$ and $\Lambda_j \in \mathbb{R}^{p \times p}$, where $p = 2n + c + d$:

$$\lambda_j := \begin{bmatrix} \alpha_j \\ \beta_j \\ u_j \\ v_j \end{bmatrix} \quad \text{and} \quad \Lambda_j := \begin{bmatrix} \Lambda_{\alpha_j \alpha_j^\top} & \Lambda_{\alpha_j \beta_j^\top} & \Lambda_{\alpha_j u_j^\top} & \Lambda_{\alpha_j v_j^\top} \\ \Lambda_{\beta_j \alpha_j^\top} & \Lambda_{\beta_j \beta_j^\top} & \Lambda_{\beta_j u_j^\top} & \Lambda_{\beta_j v_j^\top} \\ \Lambda_{u_j \alpha_j^\top} & \Lambda_{u_j \beta_j^\top} & \Lambda_{u_j u_j^\top} & \Lambda_{u_j v_j^\top} \\ \Lambda_{v_j \alpha_j^\top} & \Lambda_{v_j \beta_j^\top} & \Lambda_{v_j u_j^\top} & \Lambda_{v_j v_j^\top} \end{bmatrix} \quad \text{such that} \quad \Lambda_j = \lambda_j \lambda_j^\top. \tag{11}$$

This allows us to express the objective as a convex function of $\Lambda_j$:

$$\left\| \sum_{j=1}^m P_\alpha \Lambda_j P_v^\top - Y \right\|_F^2 + \frac{\gamma}{2} \sum_{j=1}^m \langle P_u^\top P_u + P_v^\top P_v, \Lambda_j \rangle. \tag{12}$$

Similarly, we can reformulate the constraints in Eq. (10) as

$$P_{\alpha\beta} \lambda_j \geq 0, \quad \text{trace}(P_\alpha \Lambda_j P_\beta^\top) = 0, \quad \text{and} \quad M \lambda_j = 0_{n \times 1}, \quad \text{for} \quad j = 1, \dots, m. \tag{13}$$

So far, we have shown that (NN-Train) is equivalent to the following problem:

$$\min_{\lambda_j \in \mathbb{R}^p} \left\| \sum_{j=1}^m P_\alpha \Lambda_j P_v^\top - Y \right\|_F^2 + \frac{\gamma}{2} \sum_{j=1}^m \langle P_u^\top P_u + P_v^\top P_v, \Lambda_j \rangle$$

$$\text{s.t.} \quad \text{for } j = 1, \dots, m: \qquad \qquad \text{(NN-Train-Lifted)}$$

$$M \lambda_j = 0_{n \times 1}$$

$$\text{trace}(P_\alpha \Lambda_j P_\beta^\top) = 0$$

$$\Lambda_j = \lambda_j \lambda_j^\top \quad \text{and} \quad P_{\alpha\beta} \lambda_j \geq 0_{2n \times 1}.$$

Next, we will show that (CP-NN) is also equivalent to (NN-Train-Lifted) when $m$ is sufficiently large. Since $\boldsymbol{\Lambda} \in \mathcal{PSD}$, it follows that $\mathrm{trace}(\boldsymbol{M\Lambda M}^\top) = 0$ and $\mathrm{trace}(\boldsymbol{P}_\alpha \boldsymbol{\Lambda} \boldsymbol{P}_\beta^\top) = 0$ is equivalent to $\mathrm{trace}(\boldsymbol{P}_\alpha \boldsymbol{\Lambda} \boldsymbol{P}_\beta^\top) + \mathrm{trace}(\boldsymbol{M\Lambda M}^\top) = 0$. By Lemma 2, we can find a positive integer $R$ such that any matrix $\boldsymbol{\Lambda} \in \mathcal{PSD}$ such that $\boldsymbol{P}_{\alpha\beta} \boldsymbol{\Lambda} \boldsymbol{P}_{\alpha\beta}^\top \in \mathcal{CP}$ can be factorized as

$$\boldsymbol{\Lambda} = \sum_{j=1}^R \boldsymbol{\lambda}_j \boldsymbol{\lambda}_j^\top \quad \text{for some} \quad \boldsymbol{\lambda}_j \in \mathbb{R}^p \quad \text{such that} \quad \boldsymbol{P}_{\alpha\beta} \boldsymbol{\lambda}_j \geq \boldsymbol{0}_{2n \times 1}. \tag{14}$$

Given this decomposition, we can write

$$\mathrm{trace}(\boldsymbol{M\Lambda M}^\top) = \sum_{j=1}^R \mathrm{trace}\left(\boldsymbol{M}\boldsymbol{\lambda}_j \boldsymbol{\lambda}_j^\top \boldsymbol{M}^\top\right) = \sum_{j=1}^R \|\boldsymbol{M}\boldsymbol{\lambda}_j\|^2. \tag{15}$$

Therefore, $\mathrm{trace}(\boldsymbol{M\Lambda M}^\top) = 0$ if and only if $\boldsymbol{M}\boldsymbol{\lambda}_j = \boldsymbol{0}_{n \times 1}$ for all $r = 1, \dots, R$. Similarly,

$$\mathrm{trace}(\boldsymbol{P}_\alpha \boldsymbol{\Lambda} \boldsymbol{P}_\beta^\top) = \sum_{j=1}^R \mathrm{trace}\left(\boldsymbol{P}_\alpha \boldsymbol{\lambda}_j \boldsymbol{\lambda}_j^\top \boldsymbol{P}_\beta^\top\right) = \sum_{j=1}^R \langle \boldsymbol{P}_\alpha \boldsymbol{\lambda}_j, \boldsymbol{P}_\beta \boldsymbol{\lambda}_j \rangle. \tag{16}$$

Note that $\langle \boldsymbol{P}_\alpha \boldsymbol{\lambda}_j, \boldsymbol{P}_\beta \boldsymbol{\lambda}_j \rangle \geq 0$ for all $j$ since both $\boldsymbol{P}_\alpha \boldsymbol{\lambda}_j$ and $\boldsymbol{P}_\beta \boldsymbol{\lambda}_j$ are non-negative valued. As a result, $\mathrm{trace}(\boldsymbol{P}_\alpha \boldsymbol{\Lambda} \boldsymbol{P}_\beta^\top) = 0$ if and only if $\mathrm{trace}\left(\boldsymbol{P}_\alpha \boldsymbol{\lambda}_j \boldsymbol{\lambda}_j^\top \boldsymbol{P}_\beta^\top\right) = \langle \boldsymbol{P}_\alpha \boldsymbol{\lambda}_j, \boldsymbol{P}_\beta \boldsymbol{\lambda}_j \rangle = 0$ for all $j = 1, \dots, R$. Therefore, the formulation in (CP-NN) is equivalent to the following:

$$\min_{\boldsymbol{\lambda}_j \in \mathbb{R}^p} \quad \left\|\sum_{j=1}^R \boldsymbol{P}_\alpha \boldsymbol{\Lambda}_j \boldsymbol{P}_v^\top - \boldsymbol{Y}\right\|_F^2 + \frac{\gamma}{2} \sum_{j=1}^R \langle \boldsymbol{P}_u^\top \boldsymbol{P}_u + \boldsymbol{P}_v^\top \boldsymbol{P}_v, \boldsymbol{\Lambda}_j \rangle$$

$$\text{s.t.} \quad \text{for } j = 1, \dots, R: \tag{17}$$

$$\boldsymbol{M}\boldsymbol{\lambda}_j = \boldsymbol{0}_{n \times 1}$$

$$\mathrm{trace}(\boldsymbol{P}_\alpha \boldsymbol{\Lambda}_j \boldsymbol{P}_\beta^\top) = 0$$

$$\boldsymbol{\Lambda}_j = \boldsymbol{\lambda}_j \boldsymbol{\lambda}_j^\top \quad \text{and} \quad \boldsymbol{P}_{\alpha\beta} \boldsymbol{\lambda}_j \geq \boldsymbol{0}_{2n \times 1}.$$

This concludes the proof. □

## C  PROOF OF THEOREM 2

We first introduce the following lemmas that will be required for stating the proof of Theorem 2.

**Lemma 4.** *Let $\boldsymbol{\Lambda} \in \mathbb{R}^{p \times p}$ be a positive semidefinite matrix of rank $k$, with a completely positive principal submatrix $\widehat{\boldsymbol{\Lambda}} \in \mathbb{R}^{q \times q}$ of CP-rank $K$. Without loss of generality, we assume $\widehat{\boldsymbol{\Lambda}}$ is the leading principal submatrix of $\boldsymbol{\Lambda}$ given by $\widehat{\boldsymbol{\Lambda}} = \boldsymbol{P}\boldsymbol{\Lambda}\boldsymbol{P}^\top$ with $\boldsymbol{P} = \begin{bmatrix} \boldsymbol{I}_{q \times q} & \boldsymbol{0}_{q \times (p-q)} \end{bmatrix}$. Then, there exists a probability measure $\nu$ and a corresponding random variable $\boldsymbol{\lambda}$ such that $\boldsymbol{\Lambda}$ can be decomposed as*

$$\boldsymbol{\Lambda} = \mathbb{E}_\nu[\boldsymbol{\lambda}\boldsymbol{\lambda}^\top] = \int_{\mathbb{R}^p} \boldsymbol{\lambda}\boldsymbol{\lambda}^\top d\nu(\boldsymbol{\lambda}), \tag{18}$$

*such that $\boldsymbol{\lambda} \in \mathbb{R}^p$ and $\boldsymbol{P}\boldsymbol{\lambda} \geq \boldsymbol{0}$ a.s. with respect to $\nu$.*

*Proof of Lemma 4.* Let us define the set $\Theta$ as follows:

$$\Theta := \{\boldsymbol{\Lambda} \in \mathbb{R}^{p \times p} : \boldsymbol{\Lambda} \in \mathcal{PSD}, \ \boldsymbol{P}\boldsymbol{\Lambda}\boldsymbol{P}^\top \in \mathcal{CP}\}. \tag{19}$$

By Lemma 2, we can find a positive integer $R$ such that any matrix $\boldsymbol{W} \in \Theta$ can be factorized as

$$\boldsymbol{\Lambda} = \sum_{j=1}^R \boldsymbol{\lambda}_j \boldsymbol{\lambda}_j^\top \quad \text{for some} \quad \boldsymbol{\lambda}_j \in \mathbb{R}^p \quad \text{such that} \quad \boldsymbol{P}\boldsymbol{\lambda}_j \geq \boldsymbol{0}. \tag{20}$$

Thus, by performing a scaling: $\boldsymbol{\lambda}_j \to \sigma_j \boldsymbol{\lambda}_j$ for some $\sigma_j \in [0,1]$ such that $\sum_{j=1}^{R} \sigma_j^2 = 1$, we can equivalently express $\Theta$ as follows:

$$\Theta := \left\{ \sum_{j=1}^{R} \sigma_j^2 \boldsymbol{\lambda}_j \boldsymbol{\lambda}_j^\top \ : \ \boldsymbol{\lambda}_j \in \mathbb{R}^p, \ \boldsymbol{P}\boldsymbol{\lambda}_j \geq \boldsymbol{0}, \ \sigma_j \in [0,1], \ \forall j = 1, \ldots, R, \ \text{and} \ \sum_{j=1}^{R} \sigma_j^2 = 1 \right\}. \tag{21}$$

We also define the set $\Gamma$ as follows:

$$\Gamma = \left\{ \int_{\mathbb{R}^p} \boldsymbol{\lambda}\boldsymbol{\lambda}^\top d\nu(\boldsymbol{\lambda}) \ : \ \nu \text{ is a probability measure over } \boldsymbol{\lambda}, \ \boldsymbol{P}\boldsymbol{\lambda} \geq \boldsymbol{0} \text{ a.s. with respect to } \nu \right\}. \tag{22}$$

It is clear that $\Theta \subseteq \Gamma$, because any element of $\Theta$, expressed as $\sum_{j=1}^{R} \sigma_j^2 \boldsymbol{\lambda}_j \boldsymbol{\lambda}_j^\top$, can be represented by the measure $\sum_{j=1}^{R} \sigma_j^2 \delta_{\boldsymbol{\lambda}_j \boldsymbol{\lambda}_j^\top}$, where $\delta_{\boldsymbol{\lambda}_j \boldsymbol{\lambda}_j^\top}$ is the Dirac delta measure centered at $\boldsymbol{\lambda}_j \boldsymbol{\lambda}_j^\top$. The Dirac delta measure $\delta_x$ has support at the point $x$, assigning probability 1 to sets containing $x$ and 0 otherwise.

To prove the opposite direction, that $\Gamma \subseteq \Theta$, we define an indicator function $I_\Theta(\cdot)$ given by

$$I_\Theta(\boldsymbol{\Lambda}) = \begin{cases} 0 & \text{if } \boldsymbol{\Lambda} \in \Theta \\ +\infty & \text{if } \boldsymbol{\Lambda} \notin \Theta \end{cases} \tag{23}$$

Since $\Theta$ is a closed convex set, it is clear that $I_\Theta$ is a closed convex function.

Let $\boldsymbol{\Lambda} = \int_{\mathbb{R}_p} \boldsymbol{\lambda}\boldsymbol{\lambda}^\top d\nu(\boldsymbol{\lambda}) \in \Gamma$ for some probability measure $\nu$. Applying Jensen's inequality gives the following:

$$I_\Theta(\boldsymbol{\Lambda}) = I_\Theta \left( \int_{\mathbb{R}_p} \boldsymbol{\lambda}\boldsymbol{\lambda}^\top d\nu(\boldsymbol{\lambda}) \right) \leq \int_{\mathbb{R}^p} I_\Theta(\boldsymbol{\lambda}\boldsymbol{\lambda}^\top) d\nu(\boldsymbol{\lambda}) = 0, \tag{24}$$

so $\boldsymbol{\Lambda} \in \Theta$. This completes the proof. $\qquad\square$

We recall the following standard result from measure theory without proof, as it will be used in the subsequent analysis:

**Fact 2.** *Let $f$ be an absolutely integrable function and $\nu$ be a measure defined on the measure space $(\mathbb{R}^p, \mathcal{B}^p)$. If $f \geq 0$ almost surely, then $\int_{\mathbb{R}^p} f(\boldsymbol{x}) d\nu(\boldsymbol{x}) = 0$ if and only if $f = 0$ almost surely.*

*Proof of Theorem 2.* We begin by rewriting (NN$\int$-Train) in constrained form:

$$\min_{\nu:\mathcal{B}^{d+c} \to [0,1]} \left\| \int_{\mathbb{R}^d \times \mathbb{R}^c} \boldsymbol{s}\boldsymbol{v}^\top d\nu(\boldsymbol{u},\boldsymbol{v}) - \boldsymbol{Y} \right\|_F^2 + \frac{\gamma}{2} \int_{\mathbb{R}^d \times \mathbb{R}^c} (\|\boldsymbol{u}\|_2^2 + \|\boldsymbol{v}\|_2^2) d\nu(\boldsymbol{u},\boldsymbol{v})$$
$$\text{s.t.} \quad \hat{\boldsymbol{s}}(\boldsymbol{u},\boldsymbol{v}) = \boldsymbol{X}\boldsymbol{u} \tag{25}$$
$$\boldsymbol{s}(\boldsymbol{u},\boldsymbol{v}) = (\hat{\boldsymbol{s}}(\boldsymbol{u},\boldsymbol{v}))_+.$$

We introduce non-negative variables $\boldsymbol{\alpha}(\boldsymbol{u},\boldsymbol{v}), \boldsymbol{\beta}(\boldsymbol{u},\boldsymbol{v}) \in \mathbb{R}_+^n$ that splits $\boldsymbol{s}$ and $\hat{\boldsymbol{s}}$ as follows:

$$\boldsymbol{\alpha}(\boldsymbol{u},\boldsymbol{v}) = \boldsymbol{s}(\boldsymbol{u},\boldsymbol{v}) \quad \text{and} \quad \boldsymbol{\beta}(\boldsymbol{u},\boldsymbol{v}) = \boldsymbol{s}(\boldsymbol{u},\boldsymbol{v}) - \hat{\boldsymbol{s}}(\boldsymbol{u},\boldsymbol{v}). \tag{26}$$

For the sake of brevity, we will use the following shorthand notation: $\boldsymbol{\alpha} := \boldsymbol{\alpha}(\boldsymbol{u},\boldsymbol{v})$ and $\boldsymbol{\beta} := \boldsymbol{\beta}(\boldsymbol{u},\boldsymbol{v})$. Using the equivalence described in Lemma 3, we can reformulate Eq. (25) as follows:

$$\min_{\nu:\mathcal{B}^{d+c} \to [0,1]} \left\| \int_{\mathbb{R}^d \times \mathbb{R}^c} \boldsymbol{\alpha}\boldsymbol{v}^\top d\nu(\boldsymbol{u},\boldsymbol{v}) - \boldsymbol{Y} \right\|_F^2 + \frac{\gamma}{2} \int_{\mathbb{R}^d \times \mathbb{R}^c} (\|\boldsymbol{u}\|_2^2 + \|\boldsymbol{v}\|_2^2) d\nu(\boldsymbol{u},\boldsymbol{v})$$
$$\text{s.t.} \quad \boldsymbol{X}\boldsymbol{u} - \boldsymbol{\alpha} + \boldsymbol{\beta} = \boldsymbol{0}_{n \times 1} \tag{27}$$
$$\text{diag}(\boldsymbol{\alpha}\boldsymbol{\beta}^\top) = \boldsymbol{0}_{n \times 1}.$$

Let $\nu(\boldsymbol{u},\boldsymbol{v})$ be a probability measure defined on the support of the variables $\boldsymbol{u}$ and $\boldsymbol{v}$. Then $\nu(\boldsymbol{u},\boldsymbol{v})$ represents a feasible solution to (NN$\int$-Train). Define the random vector $\boldsymbol{\lambda} := \boldsymbol{\lambda}(\boldsymbol{u},\boldsymbol{v}) \in \mathbb{R}^p$ with $p = 2n + c + d$ as $\boldsymbol{\lambda} := \boldsymbol{\lambda}(\boldsymbol{u},\boldsymbol{v}) := [\boldsymbol{\alpha}^\top \ \boldsymbol{\beta}^\top \ \boldsymbol{u}^\top \ \boldsymbol{v}^\top]^\top$. Also, let $\boldsymbol{\Lambda} := \mathbb{E}_\nu[\boldsymbol{\lambda}\boldsymbol{\lambda}^\top]$, where

the expectation is with respect to the probability measure $\nu(\boldsymbol{u}, \boldsymbol{v})$. We can represent the objective function in Eq. (27) as follows:

$$\left\| \mathbb{E}_\nu[\boldsymbol{\alpha}\boldsymbol{v}^\top] - \boldsymbol{Y} \right\|_F^2 + \frac{\gamma}{2} \mathbb{E}_\nu[\mathrm{Tr}(\boldsymbol{u}\boldsymbol{u}^\top + \boldsymbol{v}\boldsymbol{v}^\top)], \tag{28}$$

or equivalently, in terms of $\boldsymbol{\Lambda}$, as:

$$\left\| \boldsymbol{P}_\alpha \boldsymbol{\Lambda} \boldsymbol{P}_v^\top - \boldsymbol{Y} \right\|_F^2 + \frac{\gamma}{2} \langle \boldsymbol{P}_u^\top \boldsymbol{P}_u + \boldsymbol{P}_v^\top \boldsymbol{P}_v, \boldsymbol{\Lambda} \rangle, \tag{29}$$

which is exactly the objective function in (CP-NN).

Then, similar to the proof of Theorem 1, we can reformulate the constraints in Eq. (27) as

$$\boldsymbol{P}_{\alpha\beta}\boldsymbol{\lambda} \geq \boldsymbol{0}_{2n\times 1}, \quad \langle \boldsymbol{P}_\alpha \boldsymbol{\lambda}, \boldsymbol{P}_\beta \boldsymbol{\lambda} \rangle = 0, \quad \text{and} \quad \boldsymbol{M}\boldsymbol{\lambda} = \boldsymbol{0}_{n\times 1}. \tag{30}$$

Next, we will show that we can replace these with

$$\mathrm{trace}(\boldsymbol{M}\boldsymbol{\Lambda}\boldsymbol{M}^\top) = 0, \quad \mathrm{trace}(\boldsymbol{P}_\alpha \boldsymbol{\Lambda} \boldsymbol{P}_\beta^\top) = 0, \quad \boldsymbol{\Lambda} \in \mathcal{PSD}, \quad \text{and} \quad \boldsymbol{P}_{\alpha\beta} \boldsymbol{\Lambda} \boldsymbol{P}_{\alpha\beta}^\top \in \mathcal{CP}, \tag{31}$$

which is exactly the feasible set of (CP-NN).

By Lemma 4, we can find a probability measure $\nu$ and a corresponding random variable $\boldsymbol{\lambda}$ such that any matrix $\boldsymbol{\Lambda} \in \mathcal{PSD}$ such that $\boldsymbol{P}_{\alpha\beta} \boldsymbol{\Lambda} \boldsymbol{P}_{\alpha\beta}^\top \in \mathcal{CP}$ can be factorized as

$$\boldsymbol{\Lambda} = \mathbb{E}_\nu[\boldsymbol{\lambda}\boldsymbol{\lambda}^\top] = \int_{\mathbb{R}^p} \boldsymbol{\lambda}\boldsymbol{\lambda}^\top \, d\nu(\boldsymbol{\lambda}), \quad \text{such that} \quad \boldsymbol{P}_{\alpha\beta}\boldsymbol{\lambda} \geq \boldsymbol{0}_{2n\times 1} \text{ a.s. with respect to } \nu. \tag{32}$$

Given this decomposition, we can show that $\mathrm{trace}(\boldsymbol{M}\boldsymbol{\Lambda}\boldsymbol{M}^\top) = 0$ if and only if $\boldsymbol{M}\boldsymbol{\lambda} = \boldsymbol{0}_{n\times 1}$ a.s., by using Fact 2, since

$$\mathrm{trace}(\boldsymbol{M}\boldsymbol{\Lambda}\boldsymbol{M}^\top) = \int_{\mathbb{R}^p} \mathrm{trace}(\boldsymbol{M}\boldsymbol{\lambda}\boldsymbol{\lambda}^\top \boldsymbol{M}^\top) \, d\nu(\boldsymbol{\lambda}) = \int_{\mathbb{R}^p} \|\boldsymbol{M}\boldsymbol{\lambda}\|^2 \, d\nu(\boldsymbol{\lambda}). \tag{33}$$

Similarly, we can show that $\mathrm{trace}(\boldsymbol{P}_\alpha \boldsymbol{\Lambda} \boldsymbol{P}_\beta^\top) = 0$ if and only if $\langle \boldsymbol{P}_\alpha \boldsymbol{\lambda}, \boldsymbol{P}_\beta \boldsymbol{\lambda} \rangle = 0$ a.s., since

$$\mathrm{trace}(\boldsymbol{P}_\alpha \boldsymbol{\Lambda} \boldsymbol{P}_\beta^\top) = \int_{\mathbb{R}^p} \mathrm{trace}(\boldsymbol{P}_\alpha \boldsymbol{\lambda}\boldsymbol{\lambda}^\top \boldsymbol{P}_\beta^\top) \, d\nu(\boldsymbol{\lambda}) = \int_{\mathbb{R}^p} \langle \boldsymbol{P}_\alpha \boldsymbol{\lambda}, \boldsymbol{P}_\beta \boldsymbol{\lambda} \rangle \, d\nu(\boldsymbol{\lambda}), \tag{34}$$

and $\langle \boldsymbol{P}_\alpha \boldsymbol{\lambda}, \boldsymbol{P}_\beta \boldsymbol{\lambda} \rangle \geq 0$ a.s. since $\boldsymbol{P}_\alpha \boldsymbol{\lambda} \geq \boldsymbol{0}_{n\times 1}$ and $\boldsymbol{P}_\beta \boldsymbol{\lambda} \geq \boldsymbol{0}_{n\times 1}$ a.s. by Eq. (32). Therefore, the constraints in Eq. (30) imply those in Eq. (31), and the constraints in Eq. (31) imply Eq. (30) almost surely. However, since the objective in Eq. (29) depends only on $\boldsymbol{\Lambda} = \mathbb{E}_\nu[\boldsymbol{\lambda}\boldsymbol{\lambda}^\top]$, and sets of measure zero do not affect $\boldsymbol{\Lambda}$, we can omit the almost surely condition on the constraints without affecting the solution. □

## D  INTERPRETATION OF THE SOLUTION TO CP-NN

The solution to (CP-NN) can be interpreted as a moment matrix over the weights and activations with respect to the representing measure for the IWNN. Based on the factorization in Eq. (32), the solution to (CP-NN) is $\boldsymbol{\Lambda} = \mathbb{E}_\nu[\boldsymbol{\lambda}\boldsymbol{\lambda}^\top]$ for some measure $\nu$ and corresponding random variable $\boldsymbol{\lambda}$:

$$\boldsymbol{\Lambda} = \mathbb{E}_\nu \begin{bmatrix} \boldsymbol{\alpha}\boldsymbol{\alpha}^\top & \boldsymbol{\alpha}\boldsymbol{\beta}^\top & \boldsymbol{\alpha}\boldsymbol{u}^\top & \boldsymbol{\alpha}\boldsymbol{v}^\top \\ \boldsymbol{\beta}\boldsymbol{\alpha}^\top & \boldsymbol{\beta}\boldsymbol{\beta}^\top & \boldsymbol{\beta}\boldsymbol{u}^\top & \boldsymbol{\beta}\boldsymbol{v}^\top \\ \boldsymbol{u}\boldsymbol{\alpha}^\top & \boldsymbol{u}\boldsymbol{\beta}^\top & \boldsymbol{u}\boldsymbol{u}^\top & \boldsymbol{u}\boldsymbol{v}^\top \\ \boldsymbol{v}\boldsymbol{\alpha}^\top & \boldsymbol{v}\boldsymbol{\beta}^\top & \boldsymbol{v}\boldsymbol{u}^\top & \boldsymbol{v}\boldsymbol{v}^\top \end{bmatrix} \tag{35}$$

Suppose we solve (CP-NN) to obtain the solution $\boldsymbol{\Lambda}$ and consider the submatrix $\mathbb{E}_\nu[\boldsymbol{u}\boldsymbol{u}^\top]$. Using this, we can compute the expected kernel matrix $\mathbf{K}[\boldsymbol{X}_0, \boldsymbol{X}_1]$ over the measure $\nu$, where $\boldsymbol{X}_0 \in \mathbb{R}^{n_0 \times d}$ and $\boldsymbol{X}_1 \in \mathbb{R}^{n_1 \times d}$ are two input datasets of size $n_0$ and $n_1$, as $\mathbf{K}[\boldsymbol{X}_0, \boldsymbol{X}_1] = \boldsymbol{X}_0 \mathbb{E}_\nu[\boldsymbol{u}\boldsymbol{u}^\top] \boldsymbol{X}_1^\top = \mathbb{E}_\nu[(\boldsymbol{X}_0\boldsymbol{u})(\boldsymbol{X}_1\boldsymbol{u})^\top]$. It is thus possible to extract the expected kernel matrix of the infinite-width (or wide) neural network from the solution of (CP-NN), which stores the learned features.

# E    ADDITIONAL DETAILS ON NUMERICAL EXPERIMENTS

We used SGD with small step sizes and trained until convergence. We observed that this approach provides a better approximation of the global solution. Based on this observation, we tuned the learning rate (LR) and the number of iterations for each dataset as:

- *Random*: Initial LR = $10^{-5}$; # of iterations = 500K.
- *Spiral*: Initial LR = $10^{-3}$; # of iterations = 8K.
- *Iris*: Initial LR = $10^{-6}$; # of iterations = 2M.
- *Ionosphere*: Initial LR = $10^{-6}$; # of iterations = 2M for $\gamma = 0.1$ and 5M for $\gamma = 0.01$.
- *Pima Indians Diabetes*: Initial LR = $10^{-8}$; # of iterations = 5M for $\gamma = 0.1$ and 6M for $\gamma = 0.01$.
- *Bank Notes Authentication*: Initial LR = $10^{-6}$; # of iterations = 5M.
- *MNIST*: Initial LR = $10^{-7}$; # of iterations = 8M.

The SGD training curves for *Random* and *Spiral* with varying number of hidden neurons, ranging from 5 to 300, are shown in Figure 2.

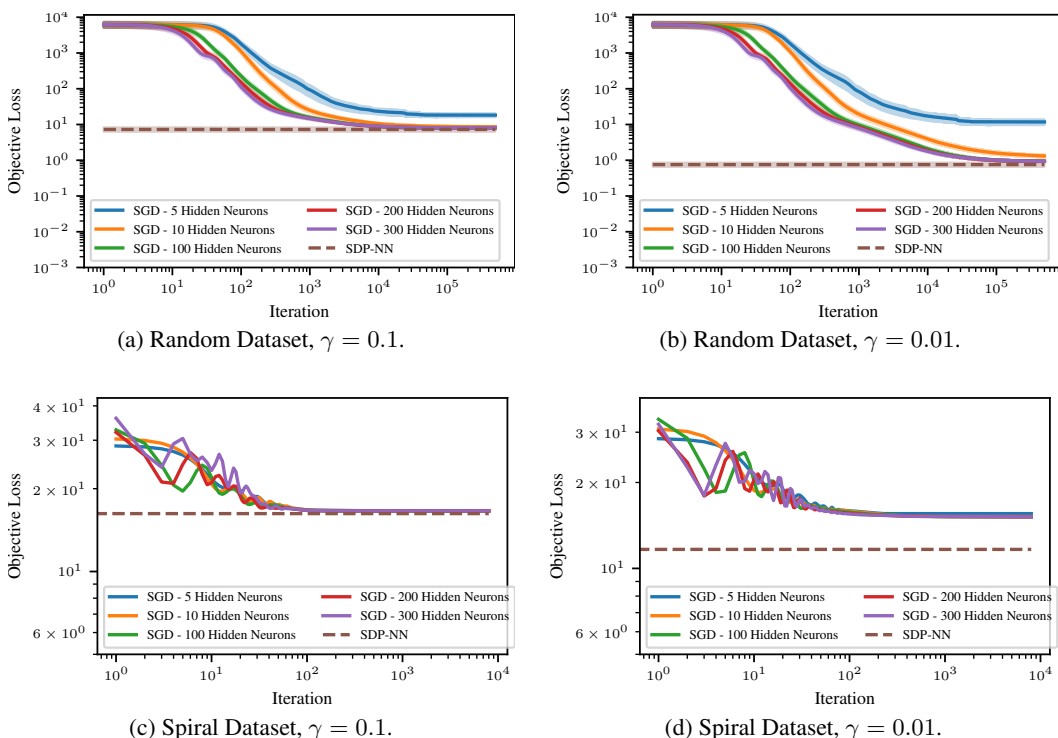

(a) Random Dataset, $\gamma = 0.1$.

(b) Random Dataset, $\gamma = 0.01$.

(c) Spiral Dataset, $\gamma = 0.1$.

(d) Spiral Dataset, $\gamma = 0.01$.

Figure 2: Objective value (training loss) for *Random* and *Spiral* datasets trained with SGD using different numbers of hidden neurons. The dashed line indicates the lower bound obtained from solving SDP-NN with the interior point method as a reference.

**Comparison with other convex relaxations.** We compared our SDP-NN approach against two convex formulations proposed in (Sahiner et al., 2021). The first is the convex semi-infinite bi-dual formulation (Equation (14) in their paper), solved using a Frank-Wolfe type algorithm, which we refer to as 'Sahiner FW.' The second is their copositive relaxation of the problem (Equation (19) in their paper), which we refer to as 'Sahiner CP,' solved with CVXPY. We used their publicly available code for these methods. The comparison was conducted on the Random dataset for regression and the Spiral dataset for classification. The results are shown in Table 4 for the regression and Table 5 for the classification. Since the provided implementation of 'Sahiner CP' was designed specifically for classification, only 'Sahiner FW' is included in Table 4.

It is important to note that the training objectives obtained by these methods are not directly comparable to SDP-NN. In particular, both 'Sahiner FW' and 'Sahiner CP' require enumerating all possible sign patterns for a given network width (300 hidden neurons in this experiment) and training dataset. As a result, they approximate the training solution with 300 hidden neurons accurately, as seen in the tables, with slightly higher objective values than SGD. In contrast, SDP-NN is a relaxation for an (in)finite-width network (and, as can be easily verified, the critical width $R$ in these examples can be larger than 300), hence provides a lower-bound on the global optimal value. It is also worth noting that the epigraph level-set parameter $t$ in Sahiner FW method, which theoretically should be optimized using bisection, is instead directly set using the SGD solution for simplicity. Including bisection would further increase the runtime for Sahiner FW.

Table 4: Comparison of convex formulations for regression with the *Random* dataset.

| Algorithm | $\gamma = 0.1$ | | | $\gamma = 0.01$ | | |
|---|---|---|---|---|---|---|
| | Train Obj. | Std Dev. | Time (s) | Train Obj. | Std Dev. | Time (s) |
| SDP-NN | 7.275 | 0.976 | 11.35 | 0.755 | 0.099 | 12.47 |
| Sahiner FW | 8.213 | 1.083 | 1341.8 | 1.099 | 0.159 | 1024.8 |
| SGD | 8.090 | 1.041 | - | 0.936 | 0.116 | - |

\*Sahiner FW and SGD are applied with 300 hidden neurons.
Train objective and time are averaged over 100 random dataset.

Table 5: Comparison of convex formulations for classification with the *Spiral* dataset.

| Algorithm | $\gamma = 0.1$ | | $\gamma = 0.01$ | |
|---|---|---|---|---|
| | Train Obj. | Time (s) | Train Obj. | Time (s) |
| SDP-NN | 16.235 | 1724.05 | 11.658 | 1062.81 |
| Sahiner FW | 16.593 | 1885.60 | 15.184 | 1911.77 |
| Sahiner CP | 16.660 | 4.75 | 15.853 | 5.07 |
| SGD | 16.593 | - | 15.159 | - |

\*Sahiner FW, Sahiner CP and SGD are applied with 300 hidden neurons.

