# OpenReview forum: "Convex Formulations for Training Two-Layer ReLU Neural Networks"
_ICLR.cc/2025/Conference — ICLR 2025 Poster_

### Official Review · Reviewer_9js6 · 2024-10-23

**Soundness:** 4
**Presentation:** 3
**Contribution:** 3
**Rating:** 6
**Confidence:** 4

**Summary:**

The paper studies 2-layer ReLU neural networks trained with MSE loss and $l_2$ regularization. The main contribution of the paper is in showing that beyond certain critical width (of at most quadratic dependence on the number of samples), the standard optimization problem is equivalent to a convex problem over a matrix of quadratic number of parameters in number of samples and input/output dimensions. The optimal values of this problem match.

The equivalence further applies to infinite-width 2-layer ReLU networks with analogical loss function. The optimal solutions of the convex problem can be used to construct optimal solutions of the infinite-width problem.

Finally, the paper proposes a computationally efficient algorithm that approximates the convex, yet copositive program. This new, semidefinite program is solvable in polynomial time and the solution of the original problem can be recovered from the solution of the SDP problem by a rounding scheme described in the paper, using existing approaches such as the three-operator splitting method.

The authors then evaluate this method as well as the tightness of the SDP relaxation on some tabular datasets of small size, demonstrating (within these datasets) that the SDP relaxation is rather tight and the practical optimization algorithm works comparably to known optimization methods, despite using number of parameters that is not theoretically guaranteed to work.

**Strengths:**

- S1: The strongest part of the paper is, in my opinion, the theoretical contribution (Theorem 1). To the best of my knowledge, this convex formulation significantly improves the number of parameters (as a function of the number of samples and input/output dimension) in the convex program from high-degree polynomial [1], to quadratic. The price the authors pay is the minimal width required to achieve this equivalence, which is also quadratic in the number of samples, as opposed to linear in previous works. Nevertheless, this neat and parameter-light reformulation is of independent theoretical interest and I believe it can be used as a tool in theoretical analyses of 2-layer NNs outside the scope of this paper. I checked the proof of the result and everything seems to be all right with only a small issue that I will discuss in typos.

- S2: The paper is overall well written, clear and interesting.

[1] Sahiner, Arda, et al. "Vector-output relu neural network problems are copositive programs: Convex analysis of two layer networks and polynomial-time algorithms." arXiv preprint arXiv:2012.13329 (2020).

**Weaknesses:**

- W1: The practical contribution of this paper is limited. Even if we ignored the fact that the practical algorithm introduced in this paper is only suitable to train 2-layer neural networks, the main issue remains -- there already are algorithms in the literature that can obtain approximate solutions of the convex re-formulations using quadratic time per-iteration in the number of samples (see for instance [1]). Moreover, the experimental evaluation is not very convincing. The tested datasets are very small so the results are not very convincing. It is known (both theoretically and empirically), that gradient-based methods work well also with widths linear in the number of samples (see for instance [2], which considers only marginally different setting). Whether your method works well with R constant (the linear dependence on $n$ already comes from lifting into the higher-dimensional space) or at least sub-quadratic in $n$ is questionable (and not addressed in the paper). Moreover, the performance is comparable with already existing and well-established optimization methods (as demonstrated in the paper).

- W2: While it is written in the paper that the algorithm scales quadratically with the number of samples (which is true in the plain CP-NN formulation), recovering a solution to the original problem actually requires cubic number of parameters and thus also time. This can be seen in equation (3), where it is clear that already the dimension of the $\lambda_j$ vectors scales with $n$ and $R$ scales quadratically with $n$, thus in total the dependence is cubic. Even if an oracle gave us *directly* a solution to CP-NN problem, recovering a solution to the original problem would still require $n^3$ number of parameters.

- W3 (minor): The computational complexities of all the involved procedures should be explicitly discussed in the paper.

**Small issues:**

- Lemma 3 is wrong, the $w$ can have positive values where $\hat W$ has negative values. A correct version of this lemma removes $\lambda$ completely from the statement and replaces the $\lambda$ in the if and only if statement with $w$. This would also be consistent with the later use of this lemma in the actual proof of Theorem 1.

- Typo: line 176 you refer to Lemma 4 but should be Lemma 2.
- Typo: In line 167 index should be also added to the $u, v$ variables.
- Typo: across the entire proof of theorem 1 including the lemmas that precede it, you use $W$ and $\Lambda$ supposedly to denote the same object. Please only use one of these (preferably $\Lambda$).

[1] Bai, Yatong, Tanmay Gautam, and Somayeh Sojoudi. "Efficient global optimization of two-layer relu networks: Quadratic-time algorithms and adversarial training." SIAM Journal on Mathematics of Data Science 5.2 (2023): 446-474.

[2] Bombari, Simone, Mohammad Hossein Amani, and Marco Mondelli. "Memorization and optimization in deep neural networks with minimum over-parameterization." Advances in Neural Information Processing Systems 35 (2022): 7628-7640.

**Questions:**

- Q1: Do you expect Theorem 1 to be applicable in theoretical analyses of 2-layer NNs in different contexts?

- Q2:  Assume an oracle would give us a solution to the CP-NN problem directly. What is the computational complexity of an algorithm that would recover a solution to the NN-TRAIN problem?

- Q3: Regarding Theorem 2, are there solutions to the NN-integral-train that are not discrete?

- Q4: What would be the training loss of the recovered NN-TRAIN solution in Table 2? I am asking to see how far off would the projection back into feasible set be from the actual solution to NN-TRAIN.

- Q5: What makes the bank notes dataset so different from the others that your method is so significantly worse than all the others, while being pretty good on all the other datasets?

**Summary:** Despite several flaws outlined above, I still think this paper meets acceptance threshold due to the novel, compact and parameter-light convex formulation of the 2-layer NN training which does not rely on enumerating all the activation patterns of the intermediate layer (as in the previous literature). This result is of theoretical interest and, though simple in nature, it is novel and insightful. Therefore, I recommend acceptance. However, I condition this opinion as well as the score on the satisfying discussion period.

**After the rebuttal:** The authors satisfactorily answered my questions and discussed my concerns. I am keeping my original score.

---

> ### Author Response · Authors · 2024-11-23
>
> We thank the reviewer for their detailed feedback and for finding our paper to be novel and insightful. We acknowledge that the reviewer is right in many of their observations. We appreciate the reviewer’s grasp of our work.
>
> **Practical contribution.** Our paper provides a simple yet novel formulation that offers a new perspective on this problem. Unlike existing approaches such as NTK/NNGP or the method proposed by Sahiner et al. (2021), our approach is built on fundamentally different techniques, providing unique insights. Specifically, compared to Sahiner et al., our formulation is more compact and eliminates the need to enumerate sign patterns. We will include [1] in the discussion of related works in the revised paper. For further discussion on the practical aspects of our method, we kindly refer the reviewer to the "General Remarks" section, above.
>
> **Experimental evaluation.** We refer the reader to the “General Remarks” for a discussion on our experimental evaluation, which mainly targets validating our theoretical contributions rather than arguing for a practical training scheme. Note that, despite its limitations, our experimental settings still involve problems that cannot be solved by the prior works such as Sahiner et al. 2021.
>
> Furthermore, we agree that it would be interesting to test the width required in rounding to evaluate the observations made in [2] within our setting. We will leave a thorough investigation of this to a future study.
>
> **Quadratic vs. cubic scaling.** This is correct—while the formulation scales with $ n^2 $ (since the rank is absorbed), storing the factors requires $ n^3 $ storage. We now clarify this in the paper.
>
> **Computational complexity.** Solving CP-NN or the associated rounding problems is NP-hard. The complexity of solving an SDP depends on the specific algorithm used. A typical interior point method for solving an SDP problem with an $n \times n$ matrix variable and $m$ constraints requires approximately $\mathcal{O}(n^3 + m^2 n^2 + m^3)$ arithmetic operations per iteration and about $\mathcal{O}(\log(1/\epsilon))$ iterations to achieve an $\epsilon$-accurate solution. We now include a discussion about the computational aspects and report the runtime (see “General Remarks”) in the main paper.
>
> **Small issues.** We fixed all of the small issues and thank the reviewer for bringing these up. In the light of these, we further polished some parts of our paper.
>
> **Questions**
>
> **Q1.** Theorem 1 is applicable for any convex loss function $\ell: \mathbb{R}^d \times \mathbb{R}^c \to \mathbb{R}$ (with the corresponding change in the objective function), since our analysis focuses on the feasible sets and is largely independent of the loss function.
>
> **Q2.** This corresponds to the complexity of the rounding problem, which is a specific instance of nonnegative matrix factorization, a problem that is generally NP-hard, similar to solving the CP-NN.
>
> **Q3.** There can be solutions to NN-integral-train that are not discrete. However, for any such solution, there always exists another solution (that achieves the same objective value) with discrete measure supported on $R$ elements, where $R$ is the critical width. In Lemma 4 (which is used in Theorem 2), we observe that $\Lambda$ can be decomposed using a continuous probability measure, as a continuous probability distribution can always be constructed to match the first and second moments of a discrete distribution. Here, $\mathbb{E}[\lambda\lambda^\top]$ represents the second moment matrix of the random variable $\lambda$.
>
> **Q4.** We thank the reviewer for this insightful question. The rounding process effectively captures the main alignments, resulting in strong training and test accuracies, as noted in the paper. However, upon further investigation based on the reviewer’s suggestion, we noticed that it introduces a scaling issue, which negatively affects the loss values and renders them suboptimal. We will further explore the rounding procedure to address this issue and develop more robust solutions. We thank the reviewer for this suggestion which also poses a promising future direction.
>
> **Q5.** This is likely attributable to the problem size. We used the SCS solver in CVXPY for larger datasets. While this first-order algorithm is more scalable, it yields solutions with limited accuracy compared to interior-point methods. We believe that solving the problem with higher accuracy could resolve the discrepancy. However, we cannot currently rule out the possibility that the relaxation gap depends on the problem size, as we lack sufficient data to refute this hypothesis.

---

> > ### Comment · Reviewer_9js6 · 2024-11-25
> > **Thank you for your answers**
> >
> > Thank you for your answers. I generally found them satisfactory and honest. I will keep my score, but please include the discussion on the following items also in the paper: Quadratic vs. cubic scaling; Computational complexity; Q2; Q4; Q5.
> >
> > Regarding Q1, what I meant was, whether you believe that your loss formulation could be used as a theoretical tool in analysis of NNs in other contexts, for instance in proving generalization bounds for optimal solutions of NN-TRAIN directly utilizing its connection with CP-NN.

---

> > > ### Author Response · Authors · 2024-11-25
> > >
> > > We thank the reviewer for their positive comments and suggestions. We will certainly update the paper with all the aforementioned discussion.
> > >
> > > We share the reviewer’s enthusiasm for exploring the use of our CP formulation to bound the generalization error, as it indeed represents a fascinating future direction. While we do not currently have a direct way to make this connection, we are optimistic about its feasibility. It is well-established that generalizing solutions tend to reside in the "wide minima" of the loss landscape, implying a relationship between the generalization gap and the location of critical points. For our lifted convex formulations, the critical points are readily identifiable and can be mapped back to the non-convex landscape of the original problem through smooth parameterizations, as demonstrated by Levin et al. [*]. By characterizing the local convexity of these points, we can potentially determine which solutions generalize better.
> > >
> > > We agree that this research direction holds significant promise and plan to pursue a more detailed investigation in future work.
> > >
> > > [*] Levin, E., Kileel, J., and Boumal, N. The effect of smooth parametrizations on nonconvex optimization landscapes. Mathematical Programming. 2024.

---

> > > > ### Comment · Reviewer_9js6 · 2024-11-25
> > > >
> > > > Thank you, that sounds good.

---

### Official Review · Reviewer_T7kD · 2024-11-03

**Soundness:** 3
**Presentation:** 3
**Contribution:** 2
**Rating:** 6
**Confidence:** 3

**Summary:**

This paper proposes a novel training method for two-layer neural networks based on a formulation of a convex completely positive program. The authors also propose a semidefinite relaxation which ensures that the problem can be solved in polynomial time. Experiments on several simple synthetic and real datasets are conducted to demonstrate the performance of the proposed method in comparison to the Neural Network Gaussian Process and Neural Tangent Kernel methods.

**Strengths:**

- The idea of the proposed method based on convex completely positive program and semidefinite relaxation is interesting.

- The presentation of the paper is fairly clear.

**Weaknesses:**

- The proposed method is only for training wide two-layer neural networks with ReLU activation function. It seems that extending the method to deeper networks is non-trivial, which limits the practical value of the proposed method. It would be beneficial if the authors could discuss the possibility to implement the proposed methods to train modern deep neural networks.

- As the authors have commented, the problem (Cp-Nn) is NP-hard due to the complete positivity constraint, and as far as I can see, the corresponding (Sdp-Nn) is not guaranteed to find the actual solution to (Cp-Nn). It would be helpful if the authors can add discussions about this point.

- To my knowledge, the Neural Network Gaussian Process and Neural Tangent Kernel methods are also settings (mainly about parameter/initialization scale and network width) under which SGD training of neural networks is almost equivalent to a convex optimization procedure. It is widely agreed that NNGP and NTK cannot fully explain the success of deep learning. These models only characterize scenarios where the obtained neural networks have weights very close to their random initialization. As a result, they fail to explain how deep networks learn effective feature representations. Therefore, even if the proposed method achieves competitive results compared with NNGP and NTK, it does not necessarily mean that the proposed method can compete with actual neural network training beyond the “NTK regime”.

- The experiments are not representative enough. All datasets considered in this paper are very small datasets, and the performance of the proposed method on these datasets are insufficient to demonstrate its advantage. I think the authors should at least include results on datasets such as MNIST and CIFAR-10.

- The experiment setup requires some clarification. As mentioned above, NNGP and NTK represent specific regimes in which SGD training of neural networks may fall [1,2,3,4]. Therefore, it is not reasonable to directly list “SGD”, “NNGP”, “NTK” as different training methods. Discussion how the SGD setting differs from NNGP and NTK and why they are not equivalent according to [1,2,3,4] is necessary to explain the experiment setting clearly.

- The authors should also discuss the relationship between the proposed method and the study in [5].


[1] Simon Du, Xiyu Zhai, Barnabas Poczos, and Aarti Singh. Gradient descent provably optimizes over-parameterized neural networks. ICLR 2019.

[2] Zeyuan Allen-Zhu, Yuanzhi Li, and Zhao Song. A convergence theory for deep learning via over-parameterization. ICML 2019.

[3] Simon Du, Jason Lee, Haochuan Li, Liwei Wang, and Xiyu Zhai. Gradient descent finds global minima of deep neural networks. ICML 2019.

[4] Difan Zou, Yuan Cao, Dongruo Zhou, and Quanquan Gu. Gradient descent optimizes over-parameterized deep ReLU networks." Machine learning 2020

[5] Cong Fang, Yihong Gu, Weizhong Zhang, and Tong Zhang. Convex formulation of overparameterized deep neural networks. IEEE Transactions on Information Theory 2022

**Questions:**

I suggest that the authors should address the weaknesses point out above.

---

> ### Author Response · Authors · 2024-11-20
>
> We appreciate that the reviewer finds our proposed method interesting and clearly presented. Below, we address all the weaknesses pointed out and kindly ask the reviewer to reconsider their rating.
>
> **On practicality and empirical evaluations.** We notice that the reviewer appears to have overlooked the insights convex formulations can offer. We contribute to a deeper understanding of neural networks. We believe this to be a valuable goal, even if it does not immediately translate to a novel intervention to better train neural networks bringing performance advantages. For overall concerns regarding empirical evaluations and the practical value of our contributions, we refer the reviewer to our “General Remarks”, where we have thoroughly discussed these aspects.
>
> **NTK, NNGP and SGD.** We would like to clarify a few potential misunderstandings here. **Firstly**, our aim is not to propose CP-NN or SDP-NN as replacements for actual neural network training, nor to position them as direct competitors to NNGP or NTK. Instead, the primary objective of this work is to introduce a novel perspective by framing neural network training within a convex optimization framework, which is grounded in fundamentally different principles compared to kernel-based methods. Consequently, whether or not the method goes beyond the NTK regime is outside the scope here.
>
> **Second**, we agree with the reviewer that NTK/NNGP cannot fully explain the success of deep learning due to their inherent limitations. Yet, unlike the reviewer suggested, our formulation differs significantly and does not share the same constraints. Unlike NTK and NNGP, **our convex formulation is exact** and applies to **both finite and infinite-width regimes** as long as the critical-width threshold is satisfied. Additionally, **our method is not dependent on initialization**. Any empirical inexactness arises solely from the relaxation and rounding processes, which do not undermine the theoretical validity of our contributions, even if the computation is NP-hard. As such, drawing theoretical conclusions solely based on empirical performance would overlook the broader implications and insights brought by our work.
>
> **Third**, we would like to clarify the distinctions between NNGP, NTK, and SGD. NNGP models the prior predictive distribution of an untrained neural network in the infinite-width limit, while NTK captures the training dynamics of gradient descent under similar assumptions. Neither of these frameworks is equivalent to SGD. While NTK can approximate SGD dynamics under specific overparameterized conditions (as noted in the references provided by the reviewer), this correspondence is not universal and relies on several assumptions. Importantly, as stated in the paper, we use SGD with a hidden layer of 300 neurons in our experiments. Under these conditions, we do not expect SGD to align with NTK.
>
> We now include this discussion into our paper. We hope these clarifications address the reviewer’s concerns and highlight our distinct contributions.
>
> **Fang et al. 2022.** Thank you for bringing this reference [5] to our attention; we have included it in the related works section. However, the techniques employed in that work and ours are entirely different. While they focus on deep network models, they particularly assume activation functions with continuously differentiable gradients, which makes their approach **not applicable to ReLU neural networks**. Moreover, their convex representation for infinite width neural network training is infinite-dimensional, with a discrete model obtained through a sampling approach that guarantees equivalence only in the limit as the number of neurons approaches infinity. In contrast, our work focuses on two-layer ReLU neural networks, characterizes a finite critical width, and introduces a finite-dimensional convex optimization formulation for (in)finite width neural network training.

---

> > ### Comment · Reviewer_T7kD · 2024-11-25
> >
> > I have read the other reviewers' comments. the authors' responses and general remarks. I appreciate the authors' detailed responses, but I have to say I am not fully convinced.
> >
> > (1) The authors' general remarks confirmed that MNIST and CIFAR-10 exceed the current computational capacity. While I appreciate the authors' honest comment, I think this demonstrates an important weakness of the proposed method.
> >
> > (2) I would like to clarify that, my original comment "NTK/NNGP cannot fully explain the success of deep learning due to their inherent limitations." was not saying that the proposed methods in this paper necessarily share the same constraints. What I meant is that, only demonstrating the proposed method gives comparable performance to NTK/NNGP is insufficient.
> >
> > (3) For the same reason, only demonstrating that the proposed method has advantage over NTK/NNGP is insufficient either. Moreover, I am still not fully convinced that the proposed method has advantage over NTK/NNGP. Although the authors provided some advantages of the proposed method in the response, I believe there are more important disadvantages:
> >
> > 1. NTK/NNGP establishes guarantees of an practical algorithm, (stochastic) gradient descent, in deep learning, which is exactly what people use in practice.
> >
> > 2. NTK/NNGP theory covers deep neural networks
> >
> > 3. NTK/NNGP theory can be verified on benchmark datasets such as MNIST, CIFAR-10.
> >
> > (4) I think it would greatly help clarify the significance of this paper if the authors could add a detailed discussion on how the analysis provides insights into deep neural networks. Moreover, it would be extremely helpful if the authors could propose practical variants of the proposed algorthms on MNIST/CIFAR-10, even if these practical variants may not enjoy the same theoretical guarantees.

---

> ### Author Response · Authors · 2024-11-27
>
> We thank the reviewer for expressing further concerns. We must admit that we suspect the reviewer might be overlooking the main point in our paper in favor of prioritizing performance or comparing to NTK. This is further evident from the reviewer's repeated emphasis on MNIST/CIFAR benchmarks, which are themselves considered to be toy in the regime of modern machine learning.
>
> Please note that the inability to handle the large problem sizes, as in MNIST/CIFAR, is not a unique limitation of our method but common to all similar works within the prominent convex-formulations literature, to which our paper contributes. We would like to reiterate that while our method scales to larger problem sizes compared to prior work, such as Sahiner et al. 2021, improving training performance is not the primary objective of our research. Therefore, it would be inappropriate to assess our work solely on this criterion. Instead, our experiments are a testament to the theoretical correctness of our approach, and they should not be interpreted as an attempt to provide new training algorithms. Our contribution lies in offering a novel perspective and advancing the theoretical understanding of neural networks through convex formulations.
>
> We also wish to stress that our theory is fundamentally distinct than NTK (as we explained) and we do not see the need to contrast them. Nevertheless, our contributions are advantageous over NTK due to:
>
>   (i) **Finite networks.** Thanks to the critical width, we can **theoretically work with finite neural networks** without resorting to approximations or truncation. This is a significant step beyond the limitations of NTK/NNGP.
>
>   (ii) **Initialization-independence.** Our convex formulation does not assume a particular initialization and provides **an exact closed form**. NTK/NNGP are strongly tied to the initialization.
>
> While not positioning ourselves as direct competitors to NTK/NNGP, we believe that these are clear theoretical advantages, that are important on several fronts we summarize below:
>
> **Advantages of our convex formulations.** Convex formulations of neural networks offer significant benefits in advancing our understanding of deep networks. In addition to our last comment to Reviewer 9js6, we now provide an expanded discussion and perspective on this:
>
> (i) Convex formulations are essential in converting pesky non-convex landscapes into interpretable ones. By doing so, they provide a clearer view of the solution space, revealing its geometric structure, and help identify properties of optimal solutions without getting trapped in suboptimal configurations.
>
> (ii) They pave the way for a rigorous theoretical analysis, including guarantees of convergence, uniqueness, and stability of solutions. This is in contrast to traditional training where guarantees are often limited or conditional on specific initialization schemes or overparameterization. We leave such analysis for a future work.
>
> (iii) Thanks to our exact co-positive formulation, studying how high-dimensional lifting of data affects separability and decision boundaries can offer insights into how neural networks learn representations and resolve ambiguities in the data.
>
> (iv) Bridging the gaps between optimization, geometry, and learning theory is fundamental to develop a modern theory for deep networks. This is what we precisely contribute towards.
>
> In the light of these, one particularly compelling avenue for future work involves leveraging our convex formulation to study generalization as pointed out by the Reviewer 9js6. Let us paraphrase our latest response below. Generalizing solutions are known to reside in the "wide minima" of the loss landscape. This gives a relationship between the generalization error and the location of critical points. For our lifted convex formulations, the critical points are readily identifiable and can be mapped back to the non-convex landscape of the original problem through smooth parameterizations, as demonstrated by Levin et al. [*]. By characterizing the local convexity of these points, we can potentially determine which solutions generalize better. While we have not yet investigated this, we plan to make such connections clearer in the future.
>
> **MNIST/CIFAR evaluations.** We have nonetheless initiated experiments on a subsampled version of MNIST, following a similar approach to Sahiner et al. (2021), who additionally applied data whitening to make their computations feasible. Notably, unlike Sahiner et al., we do not require data whitening to ensure computationally tractable sign patterns. The results of these experiments will be included in a further revision.
>
> We will include all these discussions into our paper and hope that this expanded exposition provides clarity strengthening the reviewer’s grasp of our contributions.
>
> [*] Levin, E., Kileel, J., and Boumal, N. The effect of smooth parametrizations on nonconvex optimization landscapes. Mathematical Programming. 2024.

---

> > ### Comment · Reviewer_T7kD · 2024-12-02
> >
> > Thank you for your detailed response. Most of my concerns are addressed, and I have incrased my score.

---

### Official Review · Reviewer_py4d · 2024-11-03

**Soundness:** 3
**Presentation:** 4
**Contribution:** 3
**Rating:** 6
**Confidence:** 3

**Summary:**

The paper investigates the training of shallow NN using convex optimization techinques.
It provides a brief background on Semi-Definite programming and Copositive Programming. Then it provides a formulation for the training of a NN as a CP problem (for a sufficiently wide 2-layer net with ReLU activations), which is then relaxed to make it computationally feasible.
The tightness of the relaxation is evaluated empirically.

**Strengths:**

The paper is clear, and reads well.
The supplementary material provides the code to reproduce the results.
Section 2 provides a complete yet brief background on the relevant optimization topics and concepts.

**Weaknesses:**

The paper does not consider bias terms in the linear layers.
The tightness of the proposed relaxation is evaluated only empirically.
There is no convergence guarantee for the TOS rounding step.
The rounding step is performed using the critical width, however in practice this is unfeasible.
Time complexity is not considered in the evaluations.
The empirical evaluation is performed on classification tasks, using L2 loss.

**Questions:**

I have some concerns on the empirical evaluation. MSE is not a suitable loss function for classification problems, a selection of regression tasks would have been more representative. Regardless, the hyperparameters chosen for the SGD baseline are quite strange (especially for PIMA Indians). Finally, the dataset is divided in train-test split, this penalizes models that overfit on the training data. However, if the goal is to evaluate how well a model is solving the given optimization problem, it should be evaluated on the training set.

In most cases, the addition of a bias term in the linear projections significantly improves performance, especially for 2-layer ReLU networks. In fact, running the proposed SGD baselines with the addition of a bias term improves performance across the board (gaining up to +15% accuracy). Given this, the lack of a bias term and its implications should probably be noted in the text.

I don't understand the results in Table 2. The metric is the loss, lower mean better, i.e. SDP-NN significantly improves on SDG. However the way the approximation ratio is computed and is presented in the main text implies that the SDP-NN performs worse than SGD.

Some equation are labelled instead of being numbered. I personally find this beaks the flow of the main text as it is not immediately clear that it is a reference to an equation (as most people are used to see numbers). If you want to refer to some equation by a meaningful name, you can use the name in the definition, i.e. instead of "... in (CP-NN)" consider using something like "... in the CP-formulation (equation 2)".
If you decide to keep the labels, at least try to use them to refer to the equations only, not to the concepts they are tied to. For instance in "While the case of countably infinite number of hidden neurons – represented by an infinite sum in (NN) – is captured as a special case of (NN∫) with a discrete probability measure, the (NN∫) model ..." you use (NN∫) to denote both the infinite width model concept as a whole, and its equation.

---

> ### Author Response · Authors · 2024-11-20
>
> We thank the reviewer for highlighting the clarity and reproducibility of our paper as well as the informativeness of the background we provide. In what follows, we address their concerns. In the light of these, we kindly ask the reviewer to reconsider their judgement.
>
> **Evaluation using training loss.** We thank the reviewer for bringing this to discussion. Our empirical studies evaluate both generalization performance (e.g., in Table 3 using real datasets) and optimization performance (e.g., in Table 2 using random and spiral datasets). We would like to emphasize that comparing methods based solely on training loss poses challenges, especially for real-world data, where the true global minimum of the training objective is generally unknown. While the SDP relaxation provides a lower bound on the objective and SGD finds an upper bound (converging to a local solution). This fundamental difference makes direct comparisons between their results difficult. We also refer the reviewer to “General Remarks” for a broader discussion on our empirical studies.
>
> **Choice of the loss function.** It is noteworthy that our analysis is independent of the loss. Hence, we can get the same guarantees for any convex loss function as long as we use the corresponding loss function in CP-NN as well. In the revised version, we will clarify this and will incorporate regression tasks to better align with the MSE loss.
>
> **Bias term.** The reviewer is correct that the addition of a bias term can significantly improve performance, particularly for 2-layer ReLU networks. Theoretically, the inclusion of a bias term does not alter the underlying framework, as it can be incorporated by augmenting the input data with a column of ones. While our formulation does allow for a bias term, we inadvertently omitted it in our experiments. Importantly, both the SDP relaxation and SGD baselines were evaluated on the same model—without bias terms—ensuring consistency in the comparisons. Nevertheless, we recognize that incorporating a bias term would yield more meaningful empirical results. As the reviewer suggested, we will update the text to explicitly note this limitation and its implications. Additionally, we will conduct further experiments incorporating bias terms for both methods to evaluate their impact on performance. While it is computationally demanding to rerun the experiments once again with a bias term, we will certainly include those in the revision.
>
> **On the losses in Table 2.** It is important to note that SDP-NN is a relaxation of the original problem, meaning it provides a lower bound on the objective. Therefore, its loss value is not directly comparable to SGD, which provides an upper bound by finding a feasible (but possibly suboptimal) solution. The lower loss for SDP-NN does not mean it "performs better" in the traditional sense, but rather indicates the quality of the relaxation. To evaluate the relationship between the two, we compute the approximation ratio, which quantifies how close the solution obtained by SGD is to the lower bound provided by SDP-NN. We like to point out that this is not a limitation of our method but stems from the nature of the problem – the global solution of the training being unavailable.
>
> Let us elaborate on that a little. Specifically, the exact approximation ratio is defined as:
> $$
> \mathrm{AR}_{\mathrm{exact}} = \frac{ F\_{\mathrm{relaxed}}^* }{F^*},
> $$
> where $F\_{\mathrm{relaxed}}^*$ denotes the relaxed solution. However, since we do not know the true global minimum $F^*$, we instead compute the empirical approximation ratio (AR), which provides only a lower bound on how well the SDP-NN relaxation captures the training problem:
>
> $$
> \mathrm{AR} = \frac{F^*\_{\mathrm{relaxed}}}{F^*\_{\mathrm{SGD}}}.
> $$
>
> Here, $F\_{\mathrm{SGD}}^*$ corresponds to the local minimum found by SGD, and because $F\_{\mathrm{SGD}}^* \geq F^*$, we ensure that:
> $$
> \mathrm{AR} \leq \mathrm{AR}_{\mathrm{exact}}.
> $$
> This guarantees that the reported AR provides a lower bound on the actual approximation ratio.
>
> **TOS rounding step.**  Although the rounding step lacks convergence guarantees, as noted in Remark 3, it serves as a valuable tool for validating our SDP-NN relaxation. Despite the heuristic nature of rounding, the ability to extract the trained weights of a successful neural network demonstrates that the SDP-NN solution contains the necessary information. It is important to emphasize that the rounding step is not integral to the theoretical contributions of the proposed framework and is used solely for empirical evaluations.
>
> **Equation labelling.** We thank the reviewer for highlighting this point. We have now ensured that only the equations are explicitly referenced, while the associated concepts are clearly and explicitly addressed in the text.
>
> **Runtime.** We refer the reviewer to the general response regarding runtimes.

---

> ### Comment · Area_Chair_28k8 · 2024-11-26
>
> Please check if the authors' response addresses your concerns.

---

> ### Author Response · Authors · 2024-11-27
>
> As per reviewer suggestions we have incorporated bias into our formulation and conducted additional evaluations with and without the bias terms. This led to a considerable improvement in accuracy and F1 scores on the Pima Indians and Bank Notes datasets, as reported in the table below.
>
> |                          | Bias or Not       | γ = 0.1 (Pima) | γ = 0.01 (Pima) | γ = 0.1 (Bank Notes) | γ = 0.01 (Bank Notes) |
> |--------------------------|-------------------|:-------------------------:|:--------------------------:|:----------------------:|:-----------------------:|
> | **F1 Score**             | **With Bias**    | **0.714**                   | **0.744**                   | **0.991**                | **0.985**                 |
> |                          | **Without Bias** | 0.679                   | 0.703                   | 0.930                | 0.893                 |
> | **Accuracy**       | **With Bias**    | **0.672**                   | **0.703**                   | **0.991**                | **0.980**                 |
> |                          | **Without Bias** | 0.646                   | 0.625                   | 0.860                | 0.767                 |
>
> It is worth noting that for NTK/NNGP, bias was already included (to their advantage) in the main paper. We will update the main paper to include these new results, along with results from the remaining datasets. We hope that the reviewer considers these additional results in their final evaluation.

---

> > ### Comment · Reviewer_py4d · 2024-11-27
> >
> > I appreciate you taking the time to re-run the experiments. The reported results seem to roughly align with my original experiments.
> > I am waiting for the revised version of the manuscript to make a final decision.
> > Please make sure to include the revised code in the supplementary materials.

---

> > > ### Author Response · Authors · 2024-11-27
> > >
> > > We thank the reviewer for taking the time to consider our additional evaluations and for verifying them. We are pleased to inform the reviewer that the paper has now been updated with the promised results, which are presented in Table 3. Across the remaining datasets, our results with are generally comparable and, in some cases, superior, especially in terms of accuracy. Our bias-term augmented variant is now clearly the best performing method in terms of overall test accuracy. Occasional performance drops are attributable to the rounding step, as the CP-NN formulation itself remains exact in theory.
> > >
> > > Additionally, we have provided our updated code as supplementary material. The uploaded code now includes two folders: one corresponding to the original version and another with the version augmented by bias terms. Following publication, we will further merge these versions, making bias an optional argument.
> > >
> > > We kindly ask the reviewer to take these updates into account in their final evaluation. We greatly appreciate your time and consideration in this matter.

---

### Official Review · Reviewer_FSdn · 2024-11-04

**Soundness:** 3
**Presentation:** 3
**Contribution:** 3
**Rating:** 8
**Confidence:** 2

**Summary:**

In this paper, the authors propose a novel model for infinite-width neural network training based on copositive programming. In theorems 1 and 2, the authors carefully demonstrate that the training of an RELU network of sufficient width using the MSE loss is equivalent to a certain convex copositive program. Next, the authors derive a semidefinite relaxation of the proposed copositive program and an associated rounding scheme to recover weights from the solution to the SDP. On a variety of synthetic and small real datasets, the authors demonstrate that their method recovers neural network weights that achieve smaller MSE (table 2) compared to networks trained via SGD and competitive predictive performance compared to NTK-based kernel methods.

**Strengths:**

This is an interesting paper that derives an equivalence between infinite-width RELU network training and solving a certain convex copositive program. This work contributes to the growing literature relating neural network training and convex optimization. The empirical results are also promising. Overall, I think this is an interesting work that provides valuable insight.

**Weaknesses:**

The empirical results seem somewhat weak to me. The authors acknowledge the similarity of their work to earlier work relating copositive programming and RELU network training. Although the existing methods make additional assumptions on the data distribution, a numerical comparison to prior work (e.g. the approximation ratio) would be beneficial in understanding how the proposed framework compares to earlier work in practice or further discussion on the applications of their technique.

**Questions:**

- can the authors comment on the runtime of solving the SDP?
- can the authors provide additional discussion about what kinds of insights can be gained by applying the proposed framework?

---

> ### Author Response · Authors · 2024-11-20
>
> We appreciate the reviewer’s thoughtful summary of our work and their positive feedback, reflecting a clear grasp of our contributions. Below, we address the concerns raised:
>
> **Numerical comparison to prior work.** We agree with the reviewer that numerical comparisons to prior work are relevant. Yet, the goal of our experiments is to validate our SDP relaxation and not to outperform existing benchmarks. It is because our formulations as well as the related works, are intended primarily for theoretical understanding rather than practical application. In this context, different formulations should not be viewed as rivals, as they can provide different insights, as the reviewer recognized. For more details, we refer the reviewer to the general response regarding runtimes.
>
> **Runtime.** We refer the reviewer to the general response regarding runtimes.
>
> **On the insights.** Exploring the geometry of the solution space in the convex landscape and its mapping to the original nonconvex template could yield valuable insights into how neural networks resolve data. We refer the reviewer to our general response for more details.
>
> Beyond our exposition in the general remarks (see above), the exact CP-NN formulation, combined with an analog of the results by Waldspurger and Waters on the tightness of the Pataki-Barvinok (PB) bound for factorization rank in CP-NN formulations, could provide insights into the minimal width required for ReLU neural networks to succeed. Specifically, Waldspurger and Waters studied the MaxCut SDP and its factorized (non-convex) formulation. It is well-established that the factorized problem does not exhibit spurious local minima when the factorization rank exceeds the PB bound, which scales as $O(\sqrt{n})$, where $n$ is the number of vertices (number of constraints, for a more general SDP template). Recently, Waldspurger and Waters demonstrated that the PB bound is tight, meaning that even if the original SDP problem has a unique rank-1 solution, the factorized (non-convex) problem can fail unless the factorization rank is greater than the PB bound. This result provides critical guidance for selecting the rank in matrix factorization problems.
>
> We have already touched upon these research directions in the conclusion of our main paper and we will make these connections, which motivate immediate attention to studying CP-NN formulations and their corresponding factorized (non-convex) representations, clearer.

---

### Author Response · Authors · 2024-11-20
**General Remarks**

We thank all the reviewers for their comprehensive and constructive feedback. We appreciate that the reviewers find our theoretical contribution to be strong [9js6], our formulation to be neat and parameter-light [9js6]; our paper to be interesting [FSdn,9js6, T7kD], clear [T7kD] and reproducible [py4d]; our empirical results to be promising providing valuable insights [FSdn] with an informative background on convex formulations [py4d]. We begin by addressing the common concerns below.

**Practical applicability.** We appreciate the reviewers’ concern regarding the practical applicability of our algorithm. Like many works that introduce convex formulations for neural network training, our method faces computational challenges in its current form, and we do not claim that it is intended to replace current training practices. As acknowledged by [9js6], the primary focus of this paper lies in its theoretical contributions rather than in providing a directly applicable solution for modern deep neural networks.

Our work addresses fundamental questions about the behavior and training of neural networks, contributing to the broader understanding of deep learning—a challenge shared across the research community. Specifically, we explore the geometry of the solution space within the convex landscape and its mapping to the original nonconvex formulation. To be specific, our convex formulations demonstrate how a two-layer ReLU neural network resolves data in a high-dimensional lifted space. As demonstrated by [*], which introduces new tools for analyzing connections between optimization landscapes particularly in the context of lifted formulations, the critical points of the original nonconvex training problem can be related to those of the solution space in the convex landscape.

While there is indeed an inherent gap between the two-layer ReLU networks studied in this paper (and in much of the broader literature) and the sophisticated architectures used in modern AI systems, narrowing this gap requires steady, gradual progress. In addition to the theoretical insights it offers, our work makes a tangible contribution by reducing the number of parameters in the convex formulation compared to earlier approaches, representing a meaningful step toward bridging this divide.

[*] Levin, E., Kileel, J., and Boumal, N. The effect of smooth parametrizations on nonconvex optimization landscapes. Mathematical Programming. 2024.

**Extent of empirical evaluations.** We acknowledge the limitation of using small datasets as reviewers pointed out. The size of the SDP problems for common / suggested datasets like MNIST and CIFAR-10 exceeds our current computational capacity, particularly when relying on off-the-shelf SDP solvers. We stress that this limitation is not unique to our work but is shared by prior methods that are based on convex formulations operating in a similar vein. Despite this, the datasets we use are useful in validating the proposed formulation in our experiments, which are specifically designed to evaluate the performance of the proposed SDP relaxation under a controlled setting.

It is important to note that most relevant prior methods [Sahiner et al. 2021] are based on the enumeration of sign patterns and they do not scale to the real-data examples we consider, without making further data-simplification such as whitening or sampling. Being able to handle larger problem sizes than similar works is a peculiar advantage of our formulation. We can, nevertheless, compare these methods in small datasets (e.g. Spiral and Random), and we are committed to including the result of these methods into Table 2 in the revised version. We state once again that our insights aim to deepen our understanding of neural networks, particularly their geometric and optimization properties, rather than to compete directly with existing methods in practical scenarios. We plan to address the associated computational challenges and extend the applicability of our method to larger datasets in a future work.

**Runtime.** We thank the reviewers for bringing this up. We report our runtime (in seconds) in the table below. It is important to note that runtimes depend on the specific algorithms used. For simplicity and reliability, we used off-the-shelf solvers available in CVXPY (depending on the size of the problem, Mosek and SCS), which are not necessarily the fastest or most scalable options.

| Dataset        | γ = 0.1      | γ = 0.01     |
|----------------|--------------|--------------|
| Random        | 11.46       | 14.85       |
| Spiral         | 1566.65     | 923.83      |
| Iris           | 38.98       | 171.10      |
| Ionosphere     | 4416.16     | 8150.47     |
| Pima Indians   | 12255.93    | 13563.81    |
| Bank Notes     | 102918.82   | 101952.06   |

*Note: We report the average solution time for Random dataset over 100 runs.*

We have now incorporated all these discussions and the runtime table in our main paper.

---

> ### Author Response · Authors · 2024-11-27
>
> We would like to let all reviewers know that we have updated our manuscript with the additional discussions, evaluations and changes.

---

### Meta-Review · Area_Chair_28k8 · 2024-12-19

**Metareview:**

The manuscript proposes a convex formulation for training two-layer neural networks. While the methodology seems to be only limited to simple architectures, the reviewers overall have a positive evaluation of the manuscript and find the approach interesting. Therefore, the meta-reviewer recommends acceptance of the manuscript.

**Additional Comments On Reviewer Discussion:**

The authors have addressed most of the concerns of the reviewers during the discussion phase.

---

### Decision · Program_Chairs · 2025-01-22

Accept (Poster)